



# P-model v1.0: An optimality-based light use efficiency model for simulating ecosystem gross primary production

Benjamin D. Stocker[1,2], Han Wang[3], Nicholas G. Smith[4], Sandy P. Harrison[5], Trevor F. Keenan[6,7], David Sandoval[8], Tyler Davis[8,9], and I. Colin Prentice[8]

[1]CREAF, Campus UAB, 08193 Bellaterra, Catalonia, Spain
[2]Earth System Science, Stanford University, Stanford, 94305-4216, California, USA
[3]Department of Earth System Science, Tsinghua University, Haidian, Beijing, 100084, China
[4]Department of Biological Sciences, Texas Tech University, Box 43131 Lubbock, TX 79409, USA
[5]Geography and Environmental Science, Reading University, Reading, RG6 6 AH, UK
[6]Earth and Environmental Sciences Area, Lawrence Berkeley National Lab, Berkeley, CA 94709, USA
[7]Department of Environmental Science, Policy and Management, UC Berkeley, Berkeley, CA 94720, USA
[8]AXA Chair of Biosphere and Climate Impacts, Department of Life Sciences, Imperial College London, Silwood Park Campus, Ascot, Berkshire, SL5 7PY, UK
[9]Center for Geospatial Analysis, The College of William & Mary, Williamsburg, VA, 23185, USA.

**Correspondence:** B. D. Stocker (b.stocker@creaf.uab.cat)

**Abstract.** Terrestrial photosynthesis is the basis for vegetation growth and drives the land carbon cycle. Accurately simulating gross primary production (GPP, ecosystem-level apparent photosynthesis) is key for satellite monitoring and Earth System Model predictions under climate change. While robust models exist for describing leaf-level photosynthesis, predictions diverge due to uncertain photosynthetic traits and parameters which vary on multiple spatial and temporal scales. Here, we de-

scribe and evaluate a gross primary production (GPP, photosynthesis per unit ground area) model, the P-model, that combines the Farquhar-von Caemmerer-Berry model for $C_3$ photosynthesis with an optimality principle for the carbon assimilation-transpiration trade-off, and predicts a multi-day average light use efficiency (LUE) for any climate and $C_3$ vegetation type. The model is forced here with satellite data for the fraction of absorbed photosynthetically active radiation and site-specific meteorological data and is evaluated against GPP estimates from a globally distributed network of ecosystem flux measure-

ments. Although the P-model requires relatively few inputs and prescribed parameters, the $R^2$ for predicted versus observed GPP based on the full model setup is 0.75 (8-day mean, 131 sites) – better than some state-of-the-art satellite data-driven light use efficiency models. The $R^2$ is reduced to 0.69 when not accounting for the reduction in quantum yield at low temperatures and effects of low soil moisture on LUE. The $R^2$ for the P-model-predicted LUE is 0.37 (means by site) and 0.53 (means by vegetation type). The P-model provides a simple but powerful method for predicting – rather than prescribing – light use

efficiency and simulating terrestrial photosythesis across a wide range of conditions. The model is available as an R package (*rpmodel*).





# 1 Introduction

Realistic, reliable and robust estimates of terrestrial photosynthesis are required to understand variations in the carbon cycle,
monitor forest and cropland productivity, and predict impacts of global environmental change on ecosystem function (Prentice
et al., 2015). Understanding how photosynthetic rates depend on temperature, humidity, solar radiation, $CO_2$ and soil moisture
is at the core of this challenge. Process-based Dynamic Vegetation Models (DVMs) and Earth System Models (ESMs) in use
today almost always use some form of the Farquhar-von Caemmerer-Berry (FvCB) model for $C_3$ photosynthesis (Farquhar
et al., 1980; von Caemmerer and Farquhar, 1981), in combination with stomatal conductance ($g_s$) models (Ball et al., 1987;
Leuning, 1995; Medlyn et al., 2011), that couple water and carbon fluxes at the leaf surface.

The FvCB model describes the instantaneous saturating relationship between leaf-internal $CO_2$ concentrations ($c_i$) and
assimilation ($A$), and how this relationship depends on absorbed photosynthetically active radiation (APAR). It simulates $A$ as
the minimum of a light-limited and a Rubisco-limited assimilation rate, $A_J$ and $A_C$ respectively:

$$A = \min(A_J, A_C) \tag{1}$$

Although the FvCB model is standard for leaf-scale photosynthesis, and its environmental responses at time scales of minutes
to hours, DVMs and ESMs using FvCB produce divergent results for ecosystem-level fluxes and their response to environment
at longer time scales (Rogers et al., 2017). This is due to assumptions that have to be made about photosynthetic parameters
that are not predicted by the FvCB model: stomatal conductance ($g_s$) and the maximum rates of Rubisco carboxylation ($V_{\mathrm{cmax}}$)
and electron transport ($J_{\max}$) for ribulose-1,5-bisphosphate (RuBP) regeneration, which together determine the relationship
between $c_i$ and $A$. Common approaches for determining the values of $V_{\mathrm{cmax}}$ and $J_{\max}$ in DVMs and ESMs are to prescribe
fixed values per plant functional type (PFT) and attempt to simulate the distribution of PFTs in space, or to use empirical
relationships between leaf N and $V_{\mathrm{cmax}}$ and simulate leaf N internally or prescribe it per PFT (Smith and Dukes, 2013; Rogers,
2014).

While the FvCB model describes a non-linear relationship between instantaneous assimilation and absorbed light, ecosystem
production, integrated over weeks to months, scales proportionally with absorbed photosynthetically active radiation (APAR)
(Monteith, 1972; Medlyn, 1998). This observation underlies the general light use efficiency (LUE) model which describes
ecosystem-level photosynthesis (gross primary production, GPP) as the product of APAR and LUE:

$$\mathrm{GPP} = \mathrm{PAR} \cdot \mathrm{fAPAR} \cdot \mathrm{LUE}\,, \tag{2}$$

where PAR is the incident photosynthetically active radiation and fAPAR is the fraction of PAR that is absorbed by green
tissue. The LUE model is the basis for observation-driven GPP models that use fAPAR and PAR based on remote sensing data
and combine this with different approaches for simulating LUE (Running et al., 2004; Zhang et al., 2017; Field et al., 1995),
and for some forest growth models (Landsberg and Waring, 1997). Other remote sensing data-based models (Jiang and Ryu,
2016) apply the FvCB model in combination with vegetation cover and type data and prescribed $V_{\mathrm{cmax}}$ for a set of PFTs.

Here, we describe a model, referred to as the *P-model*, that unifies the FvCB and LUE models following the theory developed
by Prentice et al. (2014) and Wang et al. (2017a). The model assumes an optimality principle that balances the C cost (per





unit of assimilation) of maintaining transpiration and carboxylation ($V_{\mathrm{cmax}}$) capacities. It thus predicts how the ratio of leaf-internal to ambient $CO_2$ ($c_i : c_a = \chi$) acclimates to the environment, given temperature ($T$), water vapour pressure deficit ($D$), atmospheric pressure ($p$) and ambient $CO_2$ concentration ($c_a$) (Prentice et al., 2014). The P-model also assumes that the photosynthetic machinery tends to coordinate $V_{\mathrm{cmax}}$ and $J_{\mathrm{max}}$ in order to operate close to the intersection of the light-limited

and Rubisco-limited assimilation rates (*Coordination Hypothesis*, Chen et al. (1993); Maire et al. (2012)) under mean daytime environmental conditions. By further assuming equality in the marginal cost and benefit of $J_{\mathrm{max}}$, daily-to-monthly average assimilation rates can then be described as fractions of absorbed PAR, i.e. as a LUE model (Eq. 2) (Wang et al., 2017a).

Thus, the P-model embodies an optimality-based theory for predicting the acclimation of leaf-level photosynthesis to its environment and for simulating LUE. In combination with prescribed PAR and remotely sensed fAPAR, it estimates GPP

across diverse environmental conditions (Wang et al., 2017a). Its prediction for acclimating photosynthetic parameters reduces the number of prescribed (and temporally fixed) values and avoids the distinction of model parameterisation by vegetation types or biomes (apart from a distinction between the $C_3$ and $C_4$ photosynthetic pathways). The P-model has a futher advantage over other data-driven GPP models ((Running et al., 2004; Zhang et al., 2017), and empirically upscaled GPP estimates (Jung et al., 2011) in that it accounts for the influence of changing $CO_2$, and that it uses first principles (rather than imposed functions)

to represent effects of $T$, $D$ and $p$ (Sect. 2). The theory underlying the P-model regarding the water-carbon tradeoff has been described by Prentice et al. (2014) and applied by Keenan et al. (2017) to simulate how changes in primary production have driven the terrestrial C sink over past decades; and by Smith et al. (2019) to explain variations in observed $V_{\mathrm{cmax}}$. Wang et al. (2017a) complemented the theory by including effects of limited electron transport capacity ($J_{\mathrm{max}}$) and predicted variations in observed $\chi$ across environmental gradients.

The purpose of this paper is to summarise the theory underlying the P-model and to provide a complete description and reference for its implementation, along with open access model code, available as an R package (*rpmodel*, https://stineb. github.io/rpmodel/). The paper also provides a comprehensive evaluation of GPP and LUE simulated by the P-model, using data from ecoystem flux measurements (FLUXNET 2015 Tier 1 dataset). The evaluation focuses on different components of variability (spatial, annual, seasonal, daily anomalies) (Secs. 4.6 - 4.1). We evaluate three P-model setups (Tab 1). We

further address uncertainties associated with the fAPAR forcing (Sect. 4.4) and the uncertainties in the evaluation data by using GPP data derived from different flux decomposition methods (Sect. 4.5). The use of continuous GPP measurements, rather than experimentally disturbed measurements, makes it challenging to assess modelled GPP under extreme environmental conditions. We therefore make a further evaluation of simulated GPP during the course of soil moisture drought events (*fLUE droughts*, Sect. 4.3).

## 2 Theory


The theory underlying the P-model has been described by Wang et al. (2017a) and the derivation of equations is given therein. It is presented here again for completeness.





## 2.1 Balancing carbon and water costs

The P-model centres around a prediction for the optimal ratio of leaf-internal to ambient $CO_2$ concentration $c_i : c_a$ (termed $\chi$) that balances the costs associated with maintaining the transpiration stream and the cost of maintaining a given carboxylation capacity. The optimal balance is achieved when the two marginal costs are equal:

$$a \, \frac{\partial (E/A)}{\partial \chi} = -b \, \frac{\partial (V_{\mathrm{cmax}}/A)}{\partial \chi} \; . \tag{3}$$

Here, $a$ and $b$ are the respective unit costs. $b$ is assumed to be constant, and $a$ to scale linearly with the temperature-dependent viscosity of water $\eta(T)$, calculated following Huber et al. (2009). Below, we introduce $\beta = b/a'$, with $a = \eta^* a'$ and $\eta^* = \eta(T)/\eta(25°C)$. The optimal $\chi$ solves the above equation. We use Fick's law (Fick, 1855) to express transpiration and assimilation as a function of stomatal conductance $g_s$:

$$E = 1.6 g_s D \tag{4}$$

and

$$A = g_s c_a (1 - \chi) \; , \tag{5}$$

and use the Rubisco-limited assimilation rate from the FvCB model:

$$A = A_C = V_{\mathrm{cmax}} \, m_C \; , \tag{6}$$

with

$$m_C = \frac{c_i - \Gamma^*}{c_i + K} \; , \tag{7}$$

where $c_i$ is given by $c_a \chi$. $K$ is the effective Michaelis-Menten coefficient for Rubisco-limited assimilation (Sect. B3), and $\Gamma^*$ is the photorespiratory compensation point in the absence of dark respiration (Sect. B1). The optimal $\chi$ can be derived as

$$\chi = \frac{\Gamma^*}{c_a} + \left(1 - \frac{\Gamma^*}{c_a}\right) \frac{\xi}{\xi + \sqrt{D}} \; , \tag{8}$$

with

$$\xi = \sqrt{\frac{\beta(K + \Gamma^*)}{1.6 \eta^*}} \; . \tag{9}$$

(See Appendix E1 for intermediate steps.) Because both terms in Eq. 3 are divided by $A$, the solution is independent of whether the Rubisco-limited rate $A_C$ or the light-limited rate $A_J$ (see below) are followed. With this prediction for $\chi$, we can use the *Coordination Hypothesis* (Chen et al., 1993; Haxeltine and Prentice, 1996; Maire et al., 2012) and the light-limited assimilation rate from the FvCB model to write

$$A_J = \varphi_0 \, I_{\mathrm{abs}} \, m \; , \tag{10}$$





with

$$m = \frac{c_i - \Gamma^*}{c_i + 2\Gamma^*} \,. \tag{11}$$

This equation has the form of a LUE model (Eq. 2) in that $A_J$ scales linearly with $I_{\text{abs}}$. Using Eqs. 9 and 8, $m$ can be expressed directly as

$$m = \frac{c_a - \Gamma^*}{c_a + 2\Gamma^* + 3\Gamma^* \sqrt{\frac{1.6\eta^* D}{\beta (K + \Gamma^*)}}} \,. \tag{12}$$

The unit cost ratio $\beta$ has been estimated by Wang et al. (2017a) based on global leaf $\delta^{13}$C data and a simplified version of the P-model (assuming $\Gamma^* = 0$ and neglecting the $J_{\text{max}}$ limitation). Here, we re-estimated $\beta$ based on the full version of the model using the same global leaf $\delta^{13}$C dataset. This is more strictly consistent with the model formulation implemented here. The value for $\beta$ used here is 146.0 (unitless). Eq. 12 provides the basis for predicting $CO_2$ assimilation rates in the form of a LUE model (Eq. 2) where LUE is a function of $T$ and $p$ (both affecting $\Gamma^*$, $K$, and $\eta^*$; see Secs. B1 and B3), $D$, and $c_a$.

The prediction of optimal $\chi$ has a number of corollaries (see Appendix C). An estimate for stomatal conductance ($g_s$) and the intrinsic water use efficiency (iWUE = $A/g_s$) directly follow from the optimal water-carbon balance (Eq. 3). By assuming $A_J = A_C$, we can further derive $V_{\text{cmax}}$, as well as dark respiration ($R_d$), which is a function of $V_{\text{cmax}}$ (see Secs. C3 and C4).

## 2.2 Introducing $J_{\text{max}}$ limitation

Eq. 10 assumes that the light response of $A$ is linear up to the coordination point. In reality, rates saturate towards high light levels because the electron transport rate $J$, necessary for the regeneration of ribulose-1,5,- bisphosphate (RuBP) tends towards a maximum $J_{\text{max}}$. To account for this effect, Eq. 10 can be modified, following the formulation by Smith (1937), using a non-rectangular hyperbola relationship between $A_J$ and $I_{\text{abs}}$ to allow for the effect of finite $J_{\text{max}}$:

$$A_J = \varphi_0 \, I_{\text{abs}} \, m \, \underbrace{\frac{1}{\sqrt{1 + \left(\frac{4 \, \varphi_0 \, I_{\text{abs}}}{J_{\text{max}}}\right)^2}}}_{L} \tag{13}$$

In this equation $A_J$ is no longer linear with respect to $I_{\text{abs}}$ and thus does not have the form of a LUE model. However, $J_{\text{max}}$ is assumed here to acclimate on longer time scales to $I_{\text{abs}}$, so that the marginal gain in assimilation $A$ per unit change in $J_{\text{max}}$ is equal to the unit cost ($c$) of maintaining $J_{\text{max}}$.

$$\frac{\partial A}{\partial J_{\text{max}}} = c \tag{14}$$

The unit cost $c$ is assumed to include the maintenance of light-harvesting complexes and various proteins involved in the electron transport chain. The cost of maintaining a given $J_{\text{max}}$ is thus assumed to scale linearly with $J_{\text{max}}$ and that this proportionality is constant ($c$ is constant). By taking the derivative of Eq. 13 with respect to $J_{\text{max}}$ and re-arranging terms (see Appendix E2 for intermediate steps), we obtain the $J_{\text{max}}$ limitation factor $L$ in Eq. 13 as:

$$L = \sqrt{1 - \left(\frac{c^*}{m}\right)^{2/3}} \,, \tag{15}$$





with $c^* = 4c$. Note that $L$ is independent of $I_{\mathrm{abs}}$. Hence, $A_J$ is again a linear function of absorbed light. The cost factor $c^*$ is estimated from published values of $J_{\mathrm{max}}{:}V_{\mathrm{cmax}} = 1.88$ at 25°C. (Kattge and Knorr, 2007) and $\chi = 0.8$ (Lloyd and Farquhar, 1994) at $c^* = 0.41$ (Wang et al., 2017a). The revised LUE model thus becomes

$$A = \varphi_0\, I_{\mathrm{abs}}\, m' , \tag{16}$$

with

$$m' = m\,\sqrt{1 - \left(\frac{c^*}{m}\right)^{2/3}} . \tag{17}$$

Wang et al (2017a) showed that this formulation of $J_{\mathrm{max}}$ costs leads to a realistic dependence of the $J_{\mathrm{max}}{:}V_{\mathrm{cmax}} =$ ratio on growth temperature.

As shown by Smith et al. (2019), an alternative approach can be used to introduce the effects of $J_{\mathrm{max}}$ limitation, replacing Eq. 13 by the more widely used one-parameter family of saturation curves following Farquhar and Wong (1984). This alternative is described in Appendix E3 and implemented as an optional method in the R package *rpmodel*.

## 3 Methods

### 3.1 The light use efficiency model

$A$ is commonly expressed in mol m$^{-2}$ s$^{-1}$. For further model description and evaluation, we refer to ecosystem-scale quantities in mass units of assimilated C and model GPP (g C m$^{-2}$ d$^{-1}$) following Eq. 2 with

$$\mathrm{fAPAR} \cdot \mathrm{PPFD} \mathrel{\widehat{=}} I_{\mathrm{abs}} \tag{18}$$
$$\mathrm{LUE} \mathrel{\widehat{=}} \varphi_0(T)\, \beta(\theta)\, m'\, M_C \tag{19}$$

Here, $M_C$ is the molar mass of carbon (12.0107 g mol$^{-1}$) to convert from molar units to mass units, and PPFD is the photosynthetic photon flux density per square metre, integrated over a day (mol m$^{-2}$ d$^{-1}$). fAPAR is unitless and integrates across the canopy, i.e., from fluxes per unit leaf area to fluxes per unit ground area. LUE is in units of g C mol$^{-1}$. The intrinsic quantum yield parameter $\varphi_0$ is modelled as temperature-dependent, and an additional (unitless) empirical soil moisture stress factor ($\beta(\theta)$) is included for modelling LUE.

### 3.1.1 Temperature dependence of the intrinsic quantum yield of photosynthesis

The temperature dependence of the intrinsic quantum yield ($\varphi_0(T)$, mol mol$^{-1}$) is modelled following the temperature dependence of the maximum quantum yield of photosystem II in light-adapted leaves, determined by Bernacchi et al. (2003) as

$$\varphi_0(T) = \frac{a_L b_L}{4}\,(0.352 + 0.022\,T - 0.00034\,T^2) \tag{20}$$





where $a_L$ is the leaf absorptance, and $b_L$ is the fraction of absorbed light that reaches photosystem II. The factor $1/4$ is
introduced here as the equation given by Bernacchi et al. (2003) applies to electron transport rather than C assimilation. Here,
$(a_L b_L/4)$ is treated as a single calibratable parameter (see Section 3.3) and is henceforth referred to as $\widehat{c_L} \equiv a_L b_L/4$. (All
calibratable parameters are thereafter indicated by a hat over the symbol.) This temperature dependence was not accounted for
in earlier P-model publications (Keenan et al., 2017; Wang et al., 2017a). To test the effect of this temperature dependence
on simulated GPP, we conducted alternative simulations, where a constant $\widehat{\varphi_0}$ was calibrated instead (Sect. 3.2). Note, that
$\varphi_0$ includes the factor $a_L$ for incomplete leaf absorbtance, which is commonly quantified separately from the quantum yield
efficiency. In other vegetation models, $a_L$ is commonly ascribed a value of 0.72-0.88 (Rogers et al., 2017). Values of $\varphi_0$
used here are accordingly lower than values for the intrinsic quantum yield reported from experimental studies (Long et al.,
1993; Singsaas et al., 2001). Furthermore, within-canopy reflection and reabsorption mean that leaf-level absorptance is not
equivalent to canopy-level absorptance, thus $\varphi_0$ should be regarded as canopy-scale *effective* value of intrinsic quantum yield.
It is treated here as a calibratable parameter, which may vary according to the fAPAR forcing data set used.

### 3.1.2   Soil moisture stress

$\beta(\theta)$ is an empirical soil moisture stress function. We use results by Stocker et al. (2018) to fit this function based on two general
patterns. First, the functional form of $\beta(\theta)$ is approximated by a quadratic expression that approaches 1 for soil moisture above
a certain threshold $\theta^*$ and held constant at 1 for soil moisture values above this threshold. Here $\theta$ is the plant-available soil
water, expressed as a fraction of field capacity, and $\theta^*$ is set to 0.6. The general form is:

$$\beta = \begin{cases} q(\theta - \theta^*)^2 + 1, & \theta \leq \theta^* \\ 1, & \theta > \theta^* \end{cases} \tag{21}$$

Second, the sensitivity of $\beta(\theta)$ to extreme soil dryness ($\theta \to 0$) is related to the mean aridity, quantified as the mean annual
ratio of actual over potential evapotranspiration (AET/PET) (Stocker et al., 2018). The decline in $\beta(\theta)$ with drying soils is
steep in dry climates and less steep in less dry climates. In equation 21, the sensitivity parameter $q$ is defined by the maximum
$\beta$ reduction at low soil moisture $\beta_0 \equiv \beta(\theta = \theta_0)$, leading to $q = (\beta_0 - 1)/(\theta^* - \theta_0)^2$. Note that $q$ has a negative value. $\beta_0$ is
modelled as a linear function of the mean aridity, :

$$\beta_0 = \widehat{a_\theta} + \widehat{b_\theta}(\text{AET/PET}) \tag{22}$$

$\widehat{a_\theta}$ and $\widehat{b_\theta}$ are treated as calibratable parameters.

Soil moisture ($\theta$), AET, and PET are simulated using the SPLASH model (Davis et al., 2017), which treats soil water storage
as a single bucket and calculates potential evapotranspiration based on Priestley and Taylor (1972). We also account for a
variable water holding capacity calculated based on soil texture and depth data from SoilGrids (Hengl et al., 2014). A detailed
description of the applied empirical functions for calculating plant-available water holding capacity from texture data is given
in Appendix D.





## 3.2 Simulation protocol

We conducted multiple simulations (Tab. 1) to investigate the dependence of model performance on alternative model setups (variable/fixed soil moisture and temperature effects), alternative choices of forcing data (fAPAR), and alternative observational target data for calibration (GPP based on different flux decompositions). Parameters ($\widehat{c_L}$, $\widehat{a_\theta}$, and $\widehat{b_\theta}$) were calibrated and evaluated against the appropriate observational data for each set of simulations separately.

The setup ORG is the P-model in its original form, as described in Wang et al. (2017a). It uses a fixed quantum efficiency

of photosynthesis ($\widehat{\varphi_0}$ is calibrated, instead of $\widehat{c_L}$), and does not account for soil moisture stress ($\beta(\theta) = 1$). Here, the model is forced with fAPAR data based on MODIS FPAR (MCD15A3H), splines of 4-daily values to daily values (see Section 3.4.1), and is calibrated against GPP data from FLUXNET 2015 based on the nighttime partitioning method (NT) (see Section 3.5.2). The simulation set BRC (Bernacchi) is identical to ORG except that $\widehat{\varphi_0}$ is allowed to vary with temperature following Bernacchi et al. (2003) and Eq. 20, and $\widehat{c_L}$ is calibrated. The full P-model setup (FULL) includes the soil moisture stress

function described above, and $\widehat{c_L}$, $\widehat{a_\theta}$, and $\widehat{b_\theta}$ are calibrated simultaneously.

All additional simulations account for both temperature and soil moisture effects. The simulation set FULL_FPARitp also uses MODIS FPAR data for fAPAR, but applies a linear interpolation to get daily values to evaluate the impact of using smoothed values from the splined data set. The simulation set FULL_EVI uses MODIS EVI (MOD13Q1), splined to daily from 8-daily data, to assess to which the degree to which model performance depends on the fAPAR forcing data. See Section

3.4.1 for more information.

All simulations were calibrated against GPP data, calculated using the nighttime flux decomposition method (Reichstein et al., 2005). Additional simulation sets FULL_DT, FULL_NTsub, and FULL_Ty were used to investigate the dependence of model performance on the choice of observational data used for calibration. We used GPP data based on the nighttime decomposition method (Reichstein et al., 2005) for FULL_NTsub, the daytime decomposition method (Lasslop et al., 2010)

for FULL_DT, and an alternative decomposition method from Wang et al. (2017a) for FULL_Ty. The Ty method estimates a *constant* monthly background respiration rate fitted to match net ecosystem exchange fluxes of $CO_2$ from measurements assuming a linear or saturating dependence of GPP on PPFD. Calibration and evaluation of FULL_DT, FULL_NTsub, and FULL_Ty are done only for sites and dates where observational data is available for all three datasets (DT, NT, and Ty), hence the distinction between FULL_NTsub and FULL.

## 3.3 Model calibration


Calibration was performed only for the model parameters determining the quantum efficiency of photosynthesis ($\widehat{\varphi_0}$ or $\widehat{c_L}$, respectively) and the dependence of the sensitivity of the soil moisture stress function on average aridity (parameters $\widehat{a_\theta}$ and $\widehat{b_\theta}$). Simulated GPP was calibrated to minimise the root mean square error (RMSE) compared to observed daily GPP (Sect. 3.5). We used Generalised Simulated Annealing from the *GenSA* R package (Yang Xiang et al., 2013) to optimise model

parameters. This algorithm is particularly suited to find global minima of non-linear objective functions in situations where there can be a large number of local minima.





**Table 1.** Model setups. The standard fAPAR data is MODIS FPAR MCD15A3H, where the original data, given at 4-day intervals, is splined to daily values. Alternative greenness forcing data are based on MODIS EVI MOD13Q1, splined (spl.) from 8-day intervals to daily, and MODIS FPAR MCD15A3H, linearly interpolated (itpl.) from 4-day intervals to daily. Standard observational GPP data, used for model calibration and evaluation, are from FLUXNET 2015, based on the nighttime flux decomposition method (NT in the table, variable `GPP_NT_VUT_REF` in FLUXNET 2015). Alternative GPP data used based on the daytime flux decomposition method (DT in the table, variable `GPP_DT_VUT_REF`), and based on an alternative method (Wang et al., 2017a) (Ty in the table). For setups ORG, BRC, FULL, FULL_FPARitp, and FULL_EVI, data used for the model calibration is from all dates where NT data are available. For setups FULL_DT, FULL_Ty, and FULL_NTsub, calibration data are from all dates where data is available for all three methods DT, NT, and Ty. Column $\varphi_0(T)$ specifies whether the temperature dependence of intrinsic quantum yield is included. Column $\beta(\theta)$ specifies whether soil moisture stress is included. Columns $\widehat{\varphi_0}$, $\widehat{c_L}$, $\widehat{a_\theta}$ and $\widehat{b_\theta}$ provide the calibrated parameter values in each simulation set.

| Setup name | fAPAR data | GPP | Calibration set | $\varphi_0(T)$ | $\beta(\theta)$ | $\widehat{\varphi_0}$ | $\widehat{c_L}$ | $\widehat{a_\theta}$ | $\widehat{b_\theta}$ |
|---|---|---|---|---|---|---|---|---|---|
| ORG | FPAR MCD15A3H, spl. | NT | NT data | no | no | 0.0492 | – | – | – |
| BRC | FPAR MCD15A3H, spl. | NT | NT data | yes | no | – | 0.0817 | – | – |
| FULL | FPAR MCD15A3H, spl. | NT | NT data | yes | yes | – | 0.0870 | 0 | 0.685 |
| NULL | FPAR MCD15A3H, spl. | NT | NT data | no | no | 0.2481* | – | – | – |
| FULL_FPARitp | FPAR MCD15A3H, itpl. | NT | NT data | yes | yes | – | 0.0846 | 0 | 0.700 |
| FULL_EVI | EVI MOD13Q1, spl. | NT | NT data | yes | yes | – | 0.1293 | 0 | 0.766 |
| FULL_DT | FPAR MCD15A3H, spl. | DT | NT, DT, Ty | yes | yes | – | 0.0891 | 0 | 0.690 |
| FULL_Ty | FPAR MCD15A3H, spl. | Ty | NT, DT, Ty | yes | yes | – | 0.0868 | 0 | 0.721 |
| FULL_NTsub | FPAR MCD15A3H, spl. | NT | NT, DT, Ty | yes | yes | – | 0.0899 | 0 | 0.690 |

The value represents the fitted LUE, corresponding to ($\varphi_0 m' M_C$) in Eq. 19.

## 3.4 Forcing data

Unstressed light use efficiency, $m'$ in Eq. 19, is simulated using monthly mean values for $T$ and $D$; temporally constant site-specific elevation (used to calculate atmospheric pressure, scaled from sea-level standard pressure of 101325 Pa); and
annually varying observed atmospheric $CO_2$ (MacFarling Meure et al., 2006), identical across sites. The choice of aggregating to monthly mean values is motivated by the time scale of Rubisco turnover, which limits the rate at which photosynthetic parameters can acclimate to changing environmental conditions (McNevin et al., 2006).

Predicted monthly LUE ($m'$) is multiplied by daily varying $I_{abs}$, and response functions $\varphi_0(T)$ and $\beta(\theta)$, driven by daily varying temperature and soil moisture. This choice is motivated by the known rapid response in stomatal conductance to drying
soils (represented by $\beta(\theta)$), and the instantaneous temperature response of the quantum yield efficiency ($\varphi_0(T)$). Simulating GPP as the product of LUE and daily varying PPFD would not be consistent with the non-linear instantaneous response of $A$ to light (Eq. 10) given the acclimation time scale of photosynthesis ((Suzuki et al., 2001; Maire et al., 2012). We therefore





evaluate simulated GPP averaged over 8-day periods. The choice of appropriate model prediction and evaluation time scales is
further discussed in Sect. 5.

### 3.4.1 fAPAR

Three alternative datasets were used as model forcing for fAPAR (MODIS FPAR splined, MODIS FPAR linearly interpolated,
and MODIS EVI, see Tab. 1). MODIS FPAR data are from the MCD15A3H, Collection 6 dataset (Myneni et al., 2015), given
at a resolution of 500 m and 4 days. The data were filtered to remove data points where clouds were present, values equal
to 1.00, and outliers (more than three times the inter-quartile range). Filtered values were replaced by the mean value for the
respective day-of-year. To obtain daily varying $I_{abs}$ (Eq. 18), two alternatives were explored. For the first, daily fAPAR values
were derived using a cubic smoothing spline (function `smooth.spline()` with parameter `spar=0.01` in R (R Core Team,
2016)). For the second, values were linearly interpolated to each day. MODIS EVI data is from the MOD13Q1, collection 6
dataset (Didan, 2015), given at a resolution of 250 m and 8 days. This data were filtered based on the summary quality control
flag, removing "cloudy" pixels. Gaps were filled and data was splined to daily values. All fAPAR data were downloaded from
Google Earth Engine using the *google_earth_engine_subsets* library (Hufkens, 2017).

### 3.4.2 Meteorological data

The meteorological forcing data are derived from the FLUXNET 2015 Tier 1 dataset (daily means), which provides data from
measurements taken and processed along with the $CO_2$ flux measurements. The photosynthetic photon flux density PPFD (mol
m$^{-2}$ d$^{-1}$) is derived from shortwave downwelling radiation as PPFD $= 60 \cdot 60 \cdot 24 \cdot 10^{-6} \, k_{EC} R_{SW}$, where $k_{EC} = 2.04 \, \mu$mol J$^{-1}$
(Meek et al., 1984), and $R_{SW}$ is incoming shortwave radiation from daily FLUXNET 2015 data (variable name `SW_IN_F`,
given in W m$^{-2}$). The factor $k_{EC}$ accounts for the energy content of $R_{SW}$ and the fraction of photosynthtically active radiation
in total short-wave radiation. Vapour pressure deficit (VPD, or $D$ in Sect. 2) is taken from daily FLUXNET 2015 data (variable
name `VPD_F`) and represents means over half-hourly periods. We use daily air temperature from the FLUXNET 2015 dataset
(variable name `T_F`), defined as the mean over half-hourly data. This is a simplification, as we are not using leaf temperature
or VPD at the leaf surface, which are more directly relevant for photosynthesis.

## 3.5 Calibration and evaluation data

### 3.5.1 Site selection

We used data from 70 sites for model calibration and 131 sites for evaluation (Fig. 1, and Tab. A1). The number of valid daily
GPP data points used for the calibration set was 160,061 and 260,284 for the evaluation set. The calibration sites were selected
based on the apparent reliability of relationships between $CO_2$ fluxes, co-located greenness data, measured soil moisture, and
meteorological variables, emerging from a previous analysis (Stocker et al., 2018). For the evaluation, we used all sites except
those in cropland and seven sites where $C_4$ vegetation dominates (AU-How, DE-Kli, FR-Gri, IT-BCi, US-Ne1, US-Ne2, and
US-Ne3).

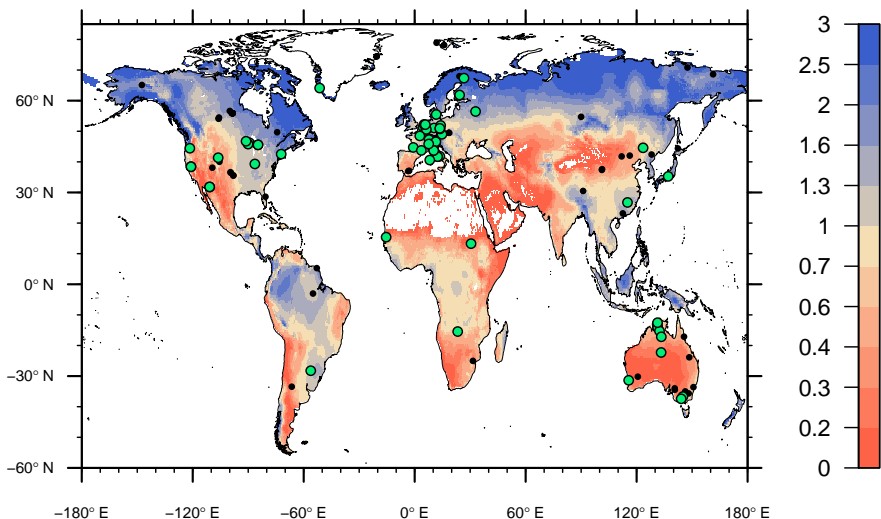

**Figure 1.** Overview of sites selected for model calibration (green dots) and evaluation (green and black dots). All sites and additional information are listed in Tab. A1. The color key represents aridity, quantified as the ratio of precipitation over potential evapotranspiration from Greve et al. (2014).

### 3.5.2 Data filtering

GPP predictions by the P-model are compared to GPP estimates from the FLUXNET 2015 Tier 1 data set (downloaded on 13 November, 2016). We used GPP based on the nighttime partitioning method (Reichstein et al., 2005) (GPP_NT_VUT_REF) and filtered out negative daily GPP values, data for which more than 50% of the half-hourly data are gap-filled and for which the daytime and nighttime partitioning methods (GPP_DT_VUT_REF and GPP_NT_VUT_REF, respectively) are inconsistent, i.e., the upper and lower 2.5% quantiles of the difference between GPP values quantified by each method. For additional

simulation sets, model calibration and evaluation was performed using GPP data based on the daytime partitioning method (GPP_DT_VUT_REF) (Lasslop et al., 2010) with analogous filtering steps, and GPP data based on an alternative method that fits a constant ecosystem respiration rate as the net ecosystem exchange under conditions where PPFD tends to zero (FULL_Ty, Wang et al. (2017a)). For all calibration and evaluation, we removed data points before the "MODIS era" (before 18th of February, 2000).

### 3.6 Evaluation methods

We evaluated both simulated LUE and GPP. The P-model (Sect. 2) predicts variations in LUE across sites (space) and months (monthly LUE = $m'$), while simulated GPP is affected by the PPFD and fAPAR data used as model forcing (Eq. 19 and Sect. 3.4). Conversely, "observed" LUE is calculated as $\text{LUE}_{\text{obs}} = \text{GPP}_{\text{obs}}/(\text{fAPAR} \cdot \text{PPFD})$ and the evaluation is thus also affected by the PPFD and fAPAR data. The evaluation of LUE tests the added explanatory power of the P-model compared to models that





**Table 2.** Description of Koeppen-Geiger climate zones (based on Falge et al. (2017)) and number of sites for which data is available per climate zone and hemisphere. Only zones with data from more than three sites are shown.

| Code | $N$ north | $N$ south | Description |
|------|-----------|-----------|-------------|
| Aw | – | 5 | Tropical savannah with dry winter |
| BSk | 5 | – | Arid steppe cold |
| Cfa | 11 | – | Warm temperate fully humid with hot summer |
| Cfb | 19 | 5 | Warm temperate fully humid with warm summer |
| Csa | 12 | – | Warm temperate with dry and hot summer |
| Csb | 4 | – | Warm temperate with dry and warm summer |
| Dfb | 17 | – | Cold fully humid warm summer |
| Dfc | 22 | – | Cold fully humid cold summer |

rely on fixed prescribed LUE values. Evaluating GPP facilitates the comparison of the model performance to similar models of terrestrial GPP. Model performance for GPP is benchmarked against a null model (NULL), which assumes a temporally constant and spatially uniform LUE. The LUE for the NULL model is fitted to observed GPP using splined MODIS FPAR and GPP data from the NT method, see Tab. 1. Thus, while LUE is constant, the NULL model preserves the spatial and temporal patterns in APAR (= fAPAR · PPFD).

### 3.6.1 Components of variability

For LUE, we separately analysed spatial (mean annual values by site) and monthly means only for the FULL setup. For GPP, we analyzed spatial, annual, seasonal (mean by day-of-year), 8-daily, and the variability in daily anomalies from the mean seasonal cycle for all setups. The seasonal variability was determined for different Koeppen-Geiger climatic zones (see Tab. 2). Information about the association of sites with climatic zones was extracted from Falge et al. (2017). Evaluations were 295 made only for climatic zones with at least five sites. For each component of variability, we calculated the adjusted coefficient of determination ($R^2_{adj}$, thereafter referred to as $R^2$), and the root mean square error (RMSE). Figures showing correlations between simulated and observed values additionally present the mean bias, the slope of the linear regression model, and the number of data points ($N$).

### 3.6.2 Drought response

The bias in GPP (modelled minus observed) was calculated for 20 days before and 80 days after the onset of a drought event as identified by Stocker et al. (2018) for 36 sites. Drought events (fLUE droughts) are periods of consecutive days where soil moisture, separated from other drivers using neural networks, reduces LUE below a given threshold. The data specifying the timing and duration of drought events was downloaded from *Zenodo* (Stocker, 2018). We then re-arranged the data to align all





**Table 3.** $R^2$ of simulated and observed GPP based on different model setups and for different components of variability.

| Setup | 8-daily | Spatial | Annual | Seasonal | var(daily) | var(annual) |
|---|---|---|---|---|---|---|
| FULL | 0.75 | 0.70 | 0.70 | 0.74 | 0.28 | 0.09 |
| BRC | 0.72 | 0.66 | 0.62 | 0.73 | 0.26 | 0.06 |
| ORG | 0.69 | 0.62 | 0.57 | 0.69 | 0.25 | 0.06 |
| NULL | 0.69 | 0.64 | 0.58 | 0.71 | 0.25 | 0.04 |
| FULL_FPARitp | 0.73 | 0.70 | 0.70 | 0.74 | 0.25 | 0.09 |
| FULL_EVI | 0.70 | 0.56 | 0.47 | 0.72 | 0.29 | 0.14 |
| FULL_DT | 0.65 | 0.69 | 0.70 | 0.65 | 0.30 | 0.08 |
| FULL_NTsub | 0.67 | 0.70 | 0.70 | 0.68 | 0.30 | 0.09 |
| FULL_Ty | 0.69 | | | 0.69 | 0.49 | |

drought events at all sites, normalised data to their median value during the ten days before the onset of droughts (normalisation

by subtracting median), and computed quantiles per day, where 'day' is defined with respect to the onset of each drought event.

## 4   Evaluation results

### 4.1   GPP variability across scales

Tables 3 and 4 provide an overview of model performance ($R^2$ and RMSE) in simulating GPP at different scales. The ORG
setup captures 69% of the variance in observed GPP with data aggregated to 8-day means (60'450 data points). Model per-
formance both with respect to explained variance ($R^2$) and the RMSE is improved by including the effects of temperature on
quantum yield efficiency in the BRC model setup ($R^2 = 72\%$), and by including the effects of soil moisture stress in the FULL
model setup ($R^2 = 75\%$, Fig. 2). Both the BRC and FULL model setup outperform the NULL model.

The $R^2$ for simulated GPP, aggregated to annual totals, ranges from 0.57 (ORG) to 0.70 (FULL). The NULL model achieves
an $R^2$ of 0.58. Most of the explanatory power of the different models for predicting annual total GPP stems from their power in
predicting between-site ("spatial") variations (Fig. 3). The $R^2$ for spatial variations ranges from 0.62 (ORG) to 0.70 (FULL),
and 0.64 for the NULL model. In contrast, inter-annual variations at a site are poorly simulated ($R^2$: 0.06-0.09 for P-model
setups, and 0.04 for the NULL model). Inter-annual variations are generally much smaller than between site variations. Thus,
capturing them is challenging. Inter-annual GPP variations are generally better simulated at sites where the variability is high
and in particular at dry sites.



**Table 4.** Root mean square error (RMSE) of simulated and observed GPP based on different model setups and for different components of variability.

| Setup | 8-daily | Spatial | Annual | Seasonal | var(daily) | var(annual) |
|---|---|---|---|---|---|---|
| FULL | 1.91 | 421.49 | 396.48 | 1.74 | 1.61 | 149.90 |
| BRC | 2.01 | 455.58 | 444.06 | 1.80 | 1.61 | 151.65 |
| ORG | 2.13 | 485.62 | 474.46 | 1.92 | 1.60 | 152.91 |
| NULL | 2.10 | 468.14 | 478.02 | 1.85 | 1.56 | 153.26 |
| FULL_FPARitp | 1.98 | 427.46 | 402.94 | 1.77 | 1.71 | 148.96 |
| FULL_EVI | 2.10 | 523.04 | 535.66 | 1.85 | 1.56 | 143.11 |
| FULL_DT | 2.08 | 390.45 | 363.55 | 1.94 | 1.73 | 155.03 |
| FULL_NTsub | 2.05 | 418.65 | 392.20 | 1.90 | 1.73 | 150.76 |
| FULL_Ty | 1.86 | | | 1.75 | 1.37 | |

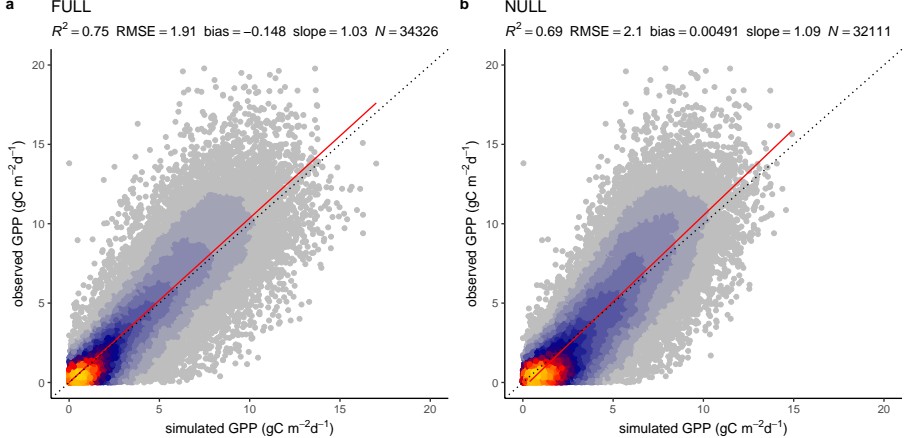

**Figure 2.** Correlation of observed and modelled GPP values of all sites pooled, mean over 8-day periods, for the model setup FULL (a) and NULL (b).

### 4.1.1 Seasonal variations

Seasonal variations are generally reliably simulated ($R^2$: 0.69-0.74 for P-model setups, and $R^2$: 0.71 for the NULL model, Fig. 4). Also the NULL model captures most of the seasonal variability, especially in climate zones Dfb and Dfc, and Cfb and Cfa. This indicates that seasonal GPP variations are largely driven by seasonal changes in insolation (PPFD) and vegetetation greenness (fAPAR). Accounting for temperature effects on the quantum yield efficiency reduces the overestimation of GPP in spring, except in the case of climate zone Dfb. Observed GPP increases are lagged compared to vegetation greenness, with a





delay of up to 2 months at some sites (e.g., US-Los). This lag is clearly visible at almost all sites in Dfb. The early season high bias is largely absent for sites in climate zone Cfb, where observed GPP starts increasing early and the simulations match the observations except at sites CZ-wet, DE-Hai, and FR-Fon, where the simulated start of season is simulated too early.

GPP is overestimated during the dry season in climate zones with a marked dry season (Aw, BSk, Csa, and Csb) in model
setups that do not account for soil moisture stress (ORG, BRC, NULL). The NULL model has the largest bias. High VPD during dry periods reduces simulated LUE and leads to lower GPP values and a smaller bias in all the P-model setups (ORG, BRC, FULL). The empirical soil moisture stress function applied in setup FULL eliminates the dryness-related bias in zones Aw, Csa, and Csb and substantially reduces this bias for sites in zone BSk. Observations suggest that GPP declines to values around zero during dry periods at sites in zone BSk (mostly savannah vegetation and grasslands, see Table A1). The remaining
bias in the FULL model, which includes the soil moisture stress function, is related to the fact that fAPAR remains relatively high and that the soil moisture stress function does not decline to zero.

The ORG and BRC models tend to underestimate peak season GPP more strongly compared to the FULL model. This is a direct consequence of the calibration which balances errors across all data points. Across-site average peak-season maximum GPP is accurately captured by the FULL model in all zones (Fig. 4), except for an underestimation of GPP in zones Aw and Cfb,
and an overestimation in zone Csa. Site-level evaluations suggests no clear relationship between peak-season underestimation and vegetation type in zone Cfb. The overestimation of peak-season GPP in zone Csa is caused by a high bias at sites with evergreen broadleaved vegetation (FR-Pue, IT-Cp2, IT-Cpz); sites with other vegetation types show no consistent peak season bias.





**Figure 3.** Correlation of modelled and observed annual GPP in simulations FULL (a), NULL (b) and FULL_EVI (c). The red line and text are based on means across years by site and represents spatial (across-site) variations. Black lines and text are based on annual values, one line for each site. Lines represent linear regressions. $R^2$ and RMSE statistics for annual values (black text) are based on pooled data from all sites. For a perfect fit between modelled and observed annual GPP values, all black lines (representing the linear regression model of annual values for a single site) would lie on the 1:1 line and have a slope of 1. Slopes that deviate substantially from 1, or even are negative, for some sites shows poor model performance in capturing inter-annual variability.



**Figure 4.** Mean seasonal cycle. Observations are given by the black line and grey band, representing the median and 33/66 % quantiles of all data (multiple sites and years) pooled by climate zone. Coloured lines represent different model setups. The annotation above each plot specifies the climate zone (see Tab. 2). Only climate zones are shown here for which data from at least five sites was available.





### 4.2 GPP target data

The different flux decomposition methods make fundamentally different assumptions regarding the sensitivity of ecosystem respiration to diurnal changes in temperature. This should lead to systematic differences in derived observational GPP values and should affect model-data disagreement.

Model predictions compare better to GPP data based on the flux decomposition method Ty (Wang et al., 2017a) than for GPP data based on the DT and NT methods. For GPP 8-day means, the model achieves an $R^2$ of 0.70 when compared to GPP
Ty (model setup FULL_Ty), as opposed to 0.65 and 0.67 compared to the DT and NT methods, respectively (FULL_DT and FULL_NTsub, Tab. 3, Fig. 7). Variations in daily anomalies are much better captured by the model when evaluating to GPP Ty ($R^2$: 0.49), as compared to evaluations against DT or NT ($R^2$: 0.30). Spatial and annual correlations are not evaluated for GPP Ty due to missing data. Correlations at the 8-day, seasonal and daily time scales rely on dates for which neither the NT, DT, nor Ty method has missing values and thus contain an equal number of data points. Therefore NT evaluations, repeated
here, are not identical to the ones above and are referred to as 'NTsub' in Tables 3 and 4.

We found a systematic low bias of simulated GPP in the peak-season in the climatic zone Cfb (warm temperate, fully humid, warm summer). However, as shown in Fig. 7, this bias does not seem to be affected by the choice of GPP evaluation data.

### 4.3 Drought response

The P-model setups that do not include the soil moisture stress function (ORG and BRC) systematically overestimate GPP
during droughts (Fig. 5). This bias increases sharply at the onset of drought events and continues to increase throughout the drought period. The bias is strongly reduced by applying the empiricial soil moisture stress function (Eq. 21) in the FULL model. A small bias remains also in the FULL model. This stems from overestimated values at a few sites (in particular AUDaP, US-Cop, US-SRG, US-SRM, US-Var, US-Whs, US-Wkg), mostly grasslands and sites in seasonally dry climate zones (Aw, BSk, and Csa, see Fig. 4), where flux measurements indicate an almost complete shut-down of photosynthetic activity during
the dry season. In contrast, the fAPAR data (MODIS FPAR) suggest values substantially greater than zero at these sites during these periods. This suggests either contributions to PAR absorption by photosynthetically inactive tissue, underestimation of LUE sensitivity to dry soils at these sites, or an overestimation of the rooting zone moisture availability by SPLASH.

### 4.4 Uncertainty from fAPAR input data

Tests of the sensitivity of model performance to alternative fAPAR forcing datasets show that the difference between splined
and linearly-interpolated MODI FPAR is negligible. However, model performance is generally better using MODIS FPAR compared to simulations using MODIS EVI. Spatial variations are well captured using MODIS FPAR (Fig. 3, $R^2$: 0.70) compared to MODIS EVI ($R^2$: 0.56). However, the $R^2$ of inter-annual variations is 0.14 for MODIS EVI and 0.09 for MODIS FPAR. In terms of biases in climate zones, the overestimation of GPP during the dry period in zone BSk is larger when using MODIS EVI than when using MODIS FPAR (Fig. 6, right). The positive spring bias in simulated GPP in zone Dfb is present
irrespective of the source of the fAPAR forcing (Fig. 6, left), as is the peak-season bias of GPP in zones BSk, Cfb, and Csb.



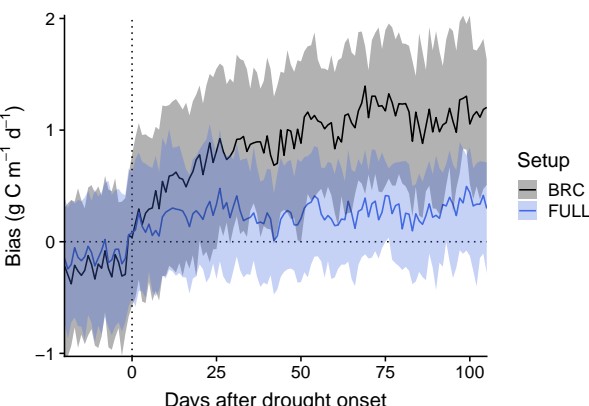

**Figure 5.** Bias in simulated GPP during the course of drought events. Simulated GPP is from a simulation with (FULL) and without (BRC) accounting for soil moisture stress. The timing of drought events is taken from Stocker et al. (2018) and is identified by an apparent soil moisture-related reduction of observed light use efficiency at 36 FLUXNET sites. The bias is calculated as simulated minus observed GPP. Data from multiple drought events and sites are aligned by the date of drought onset and aggregated across all sites and events (lines for medians, shaded ranges from the 33% and 66% quantiles).

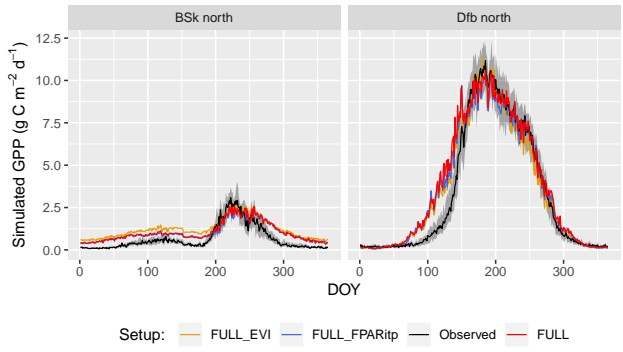

**Figure 6.** Mean seasonal cycle for model setups with different greenness forcing data for two climate zones (BSk and Dfb, both northern hemisphere). Observations are given by the black line and grey band, representing the median and 33/66 % quantiles by day-of-year (DOY) of all data (multiple sites and years) pooled by climate zone. Coloured lines represent model setups, forced with different greenness data. The annotation above each plot specifies the climate zone (see Tab. 2). Climate zones shown here are illustrative examples.

Differences between the EVI and FPAR-forced simulations depend on vegetation type. The EVI-forced simulation tends to be low biased in evergreen needle-leaved vegetation, and has generally lower values in all evergreen vegetation types compared to the FPAR-forced simulation. However, there is no general difference in model bias between simulations made with the two forcings in other vegetation types.



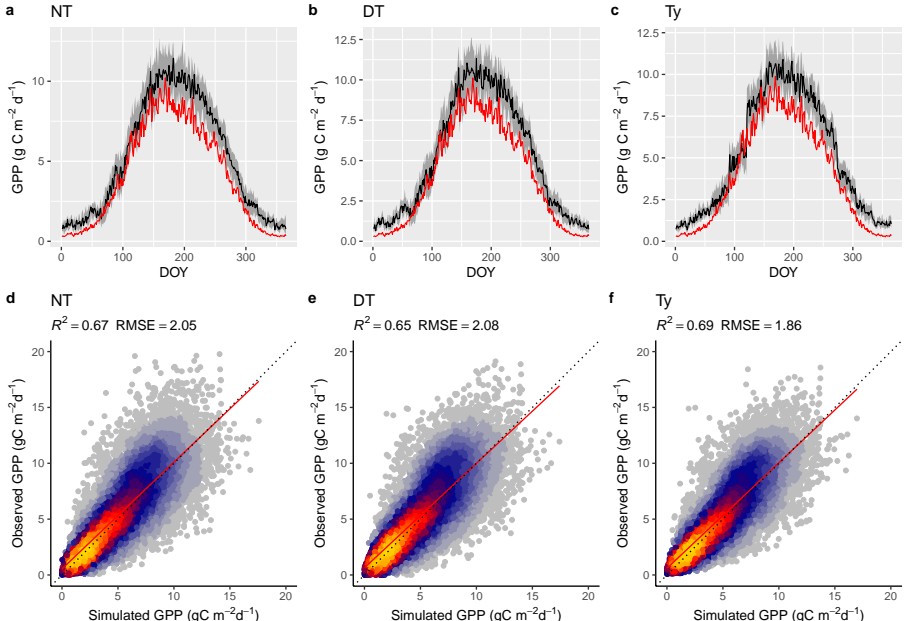

**Figure 7.** Model performance subject to comparison with different flux decomposition methods for GPP. (a-c): Mean seasonal cycle of simulated (red) and observed GPP (black) based on different flux decomposition methods. The grey band represents the 33/66 % quantiles of observed GPP by day-of-year (DOY). (d-f): Correlation of observed and simulated GPP values of all sites pooled, mean over 8-day periods. 'Observed GPP' refers to the different flux decomposition methods: DT for the daytime method (setup FULL_DT), NT for the nighttime method (setup FULL_NTsub) and Ty (setup FULL_Ty) for the method applied for data used in Wang et al. (2017b). Dotted lines in (d-f) represent the 1:1 relationship, red lines represent the fitted linear regressions.

## 4.5 Uncertainty from GPP target data

Model predictions compare better to GPP data based on the Ty than either the DT or NT methods. For GPP, 8-day means, the model achieves an $R^2$ of 0.69 when compared to GPP Ty (model setup FULL_Ty), compared to 0.65 and 0.67 for the DT and NT methods, respectively (FULL_DT and FULL_NTsub, Tab. 3, Fig. 7). Variations in daily GPP anomalies are much better captured in evaluations with Ty ($R^2$: 0.49) than DT or NT ($R^2$: 0.30). Spatial and annual correlations were not evaluated for Ty because of missing data. The systematic low bias of simulated peak-season GPP in climatic zone Cfb is not affected by the choice of GPP evaluation data (Fig. 7).

The systematic differences in the level of model-data agreement depending on the target GPP dataset (Fig. 7) reflect the fact that these datasets are derived using decomposition methods with different sensitivity to diurnal changes in temperature.





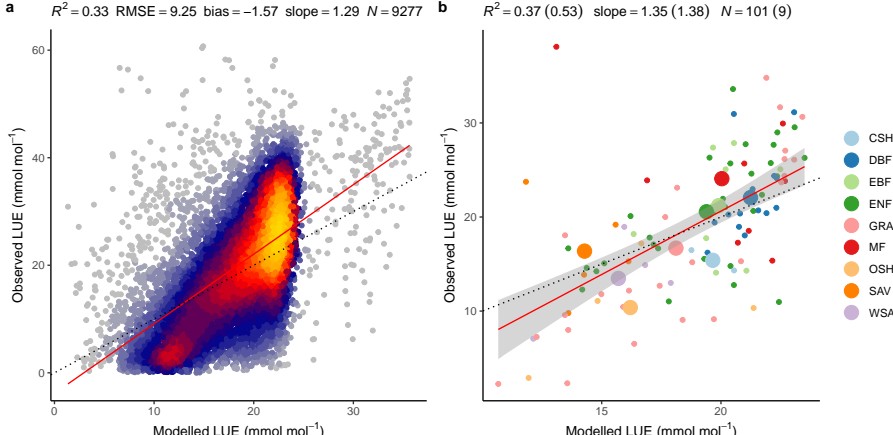

**Figure 8.** Modelled (simulations FULL) versus observed LUE. (a) Mean monthly LUE with data pooled from all sites and available years. (b) Mean annual LUE by site (small dots and color) and vegetation type (large dots and color). Model performance metrics are given at the top with numbers in brackets referring to the regression of data aggregated by vegetation types and non-bracketed numbers for data aggregated by sites. Dotted lines represent the 1:1 relationship, red lines represent the fitted linear regression to all data in (a) and to mean annual LUE by site in (b). The grey band in (b) represents the 95 % confidence interval of the linear regression. Vegetation types are: closed shrubland (CSH); deciduous broadleaf forest (DBF); evergreen broadleaf forest (EBF); evergreen needleleaf forest (ENF); grassland (GRA); mixed deciduous and evergreen needleleaf forest (MF); open shrubland (OSH); savanna ecosystem (SAV); woody savanna (WSA).

## 4.6 LUE

The FULL version of the P-model captures 37% of the variability in mean annual LUE across all sites and across the full range of observed mean annual LUE values and vegetation types (Fig. 8). 53 % of the observed LUE variation within vegetation types is captured by the model through the relationships with climate, without the need to specify parameters for specific vegetation types.

33% of the variability in monthly mean LUE is captured by the model, with data from all sites and years pooled (Fig. 8). The

model overestimates monthly LUE values and underestimates LUE at the lowest and highest end of the LUE range respectively. The low-end overestimation is reflected by the overestimation of GPP in the spring at winter-cold sites (Sect. 4.1.1) and during soil moisture droughts (Sect. 4.3). The underestimation of high monthly values is not clearly linked to any particular vegetation type.

## 5 Discussion

The performance of the P-model can be compared to results obtained from other remote-sensing driven GPP models (RS-models). In its FULL setup, the P-model achieves an $R^2$ of 0.75 and a RMSE of 1.91 g C m$^{-2}$ d$^{-1}$, in simulating 8-day mean GPP and evaluated against GPP data (NT method) from 131 sites. This can be compared to predictions from the VPM model





($R^2$: 0.74, RMSE: 2.08 g C m$^{-2}$ d$^{-1}$, 113 sites, 8-daily, Zhang et al. (2017)), or BESS ($R^2$: 0.67, RMSE: 2.58 g C m$^{-2}$ d$^{-1}$, 113 sites, 8-daily, Jiang and Ryu (2016)). The performance of the P-model in simulating *annual* GPP across all 131 sites ($R^2$:

0.70) can be compared to results from MODIS GPP (MOD17A2, $R^2$= 0.73, 12 sites, Heinsch et al. (2006), and for the updated version MOD17A2H: $R^2$= 0.62, 18 sites, Wang et al. (2017b)).

The coefficients of determination ($R^2$) of simulated versus observed values are lower for LUE (0.37 for the spatial correlation in the FULL setup, Fig. 8b) than for GPP (0.70 for the spatial correlation in the FULL setup). This is because GPP variations are strongly driven by variations in absorbed light (PPFD·fAPAR), which are observed and used for modelling. In contrast,

variations in LUE cannot be observed directly. Using remotely-sensed information for estimating LUE variations, e.g., based on sun-induced fluorescence (Frankenberg et al., 2018; Li et al., 2018; Ryu et al., 2019) or alternative reflectance indices (Gamon et al., 1992, 2016; Badgley et al., 2017), is an active field of research and the separation of remotely sensed signals into contributions by LUE and absorbed light remains challenging (Porcar-Castell et al., 2014; Ryu et al., 2019). Other remote sensing-based GPP models rely on vegetation type-specific model parameters for LUE (Zhang et al., 2017; Running et al.,

2004; Jiang and Ryu, 2016). The P-model in its FULL setup explains 53% of the variations in LUE across sites aggregated to vegetation types without relying on vegetation or biome-type specific parametrisations. In its ORG setup, it explains 21% of the variations (not shown), and 51% of the variations when excluding sites classified as 'open shrublands', which tend have a substantially lower LUE than simulated by the P-model (not shown). In spite of this substantial portion of explained variability, the NULL model with its temporally constant and spatially uniform LUE achieves higher $R^2$ values for GPP

than the ORG P-model setup at the spatial, annual, and seasonal scales (Tab. 3). This indicates that the spatial and temporal variations in absorbed light are the main drivers of GPP in LUE-type models and underlines the importance of evaluation against a NULL model benchmark. Taken together, these findings demonstrate that the P-model offers a simple but powerful method for simulating terrestrial GPP using readily available input datasets and a very small number of free (calibratable) parameters. Here, three parameters are calibrated (for the FULL setup). Other model parameters are derived from independent

field and laboratory measurements.

Accounting for the temperature-dependence of the quantum yield efficiency ($\varphi_0$) clearly improves model predictions. The parameter $\varphi_0$ is commonly treated as a constant in global vegetation models (Rogers et al., 2017). Our results indicate potential for improving DVM photosynthesis routines by accounting for the temperature-dependence of $\varphi_0$.

$\varphi_0$ appears as a linear scalar in the LUE model. However, the magnitude of this scalar is uncertain and depends on whether

incomplete light absorption by the leaf is included in the definition of $\varphi_0$ or in fAPAR data. We have used MODIS FPAR and MODIS EVI data to define fAPAR in different model setups. While the two are well correlated, their absolute values differ. Hence, we have calibrated an *apparent* quantum yield efficiency ($\widehat{\varphi_0}$) to GPP data separately for different fAPAR datasets, thereby implicitly distinguishing what components of light absorption factors are contained in the fAPAR data. The leaf absorbtance $a_L$, which is typically taken to be around 0.8 in global vegetation models (Rogers et al., 2017) is similar to

the ratio of fitted $\widehat{\varphi_0}$ values for simulation FULL and FULL_EVI, here calculated as 0.67 (Tab. 1).

An improvement in model performance is obtained by accounting for soil moisture stress using an empirical function. However, the use of an empirical function masks underlying processes. Furthermore, the use of an empirical function is not





consistent with the optimality approach that underlies the P-model. The reduced bias using an empirical soil moisture stress
function implies something is missing in the theoretical approach which rests on an assumed constancy of the unit costs of
transpiration ($a$ in Eq. 3). Prentice et al. (2014) provide a definition of $a$ that is explicit in terms of plant hydraulic traits and
physical properties that determine water transport along the plant-soil-atmosphere continuum. In particular, $a \propto (\Delta\Psi k_s)^{-1}$,
where $\Delta\Psi$ is the maximum daytime difference in leaf-to-soil water potential and $k_s$ is the sapwood area-specific permeability.
However, large variations in stomatal conductance are known to occur in response to relatively fast soil dry-downs (time scale
of days) (Keenan et al., 2010; Egea et al., 2011; Stocker et al., 2018). This suggests the potential to improve the P-model
by allowing the unit cost of transpiration to be a function of rooting-zone moisture availability, and by coupling stomatal
conductance with the soil water balance.

Observational uncertainty could affect both parameter calibration and model evaluation. Keenan et al. (2019) found a sys-
tematic bias in GPP estimates based on the nighttime partitioning method due to inhibition of leaf respiration in the light (Kok,
1949; Wehr et al., 2016), which affects fluxes unevenly throughout the season and across vegetation types. However, we found
no clear difference in model-data agreement, nor in fitted parameters, in comparisons of three alternative GPP datasets that use
different approaches to decompose net $CO_2$ exchange fluxes from eddy covariance measurements into ecosystem respiration
and GPP terms.

We have found a consistent early-season high-bias in simulated GPP for numerous sites in regions with deciduous broadleaved
vegetation in temperate and cold climates (in particular US-MMS, IT-Col, US-WCr, US-UMd, US-UMB, and US-Ha1), and
also in mixed and needle-leaved stands (in particular US-Syv, US-NR1, FI-Hyy, CA-Qfo, and CA-Man). Additional analyses
(not shown) suggested that this bias is not related to soil temperatures or the difference between soil and air temperatures in
spring. The P-model, as applied here, uses daily air temperature for simulating acclimation. Hence, a bias is to be expected
in view of the known delay in the resumption of photosynthesis after a cold dormant period (Huner et al., 1993; Oquist and
Huner, 2003; Adams et al., 2004; Verhoeven, 2014; Bowling et al., 2018). Such delays are mediated by a biochemical down-
regulation of photosynthesis, not included in the P-model, i.a. using xanthophyll carotenoids for photoprotection (Adams et al.,
2004) to balance photochemical and biochemical processes – the former being less steeply temperature-dependent (Oquist and
Huner, 2003). Cold-acclimation is a strategy to cope with frozen sap transport vessels during periods when air temperature is
already high (Bowling et al., 2018), or for protecting against frost damage during early season cold spells (Vitasse et al., 2014).
Approaches accounting for a delayed resumption of photosynthesis after cold periods (Pelkonen and Hari, 1980; Bergh et al.,
1998; Mäkelä et al., 2004) offer scope for further improvement of the P-model.

There is a positive bias in simulated GPP during the dry season at a number of sites where the vegetation phenology is
influenced by drought. The positive bias is related to the combination of using prescribed fAPAR data, which shows substantial
absorption by non-green vegetation, and insufficient sensitivity of simulated LUE to soil drying. However, GPP is accurately
simulated at other sites affected by seasonally recurring water stress. The modelled sensitivity to dry soils is determined by
the soil moisture stress function, which depends on the mean aridity of the site as estimated using a fixed depth soil moisture
"bucket". Accounting for variability in rooting zone depth, which may also be influenced by local topographical factors and
access to groundwater (Fan et al., 2013, 2017) may help to minimise model biases in drought-prone areas.



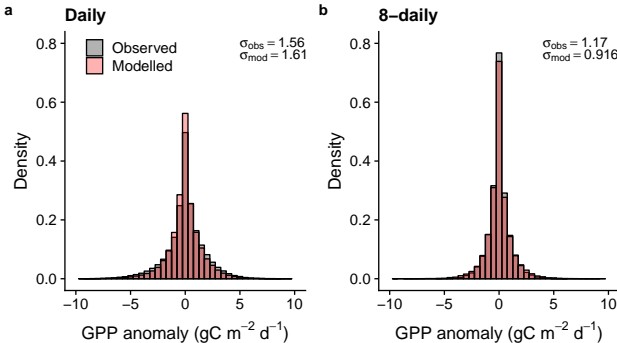

**Figure 9.** Distribution of anomalies from the mean seasonal cycle, evaluated for daily values (a) and 8-day means (b).

The current implementation of the P-model involves some simplifications in terms of climate drivers by using average daily meteorological conditions, measured above the canopy, as input. Optimality in balancing carbon and water costs for average
daily conditions is not necessarily equivalent to optimality in balancing integrated water and carbon costs over the diurnal cycle. Large variations in ambient conditions over a diurnal cycle, combined with a non-linear dependence of costs on these conditions suggest that the approach of taking average daily conditions may be an over-simplification. Nevertheless, prior evaluations have shown robust and accurate predictions of optimal $\chi$ across a range conditions (Wang et al., 2017a). Using above-canopy VPD values instead of VPD at the leaf surface for scaling water losses implicitly assumes a perfectly coupled
atmospheric boundary layer. Using above-canopy air temperature instead of leaf temperatures introduces a bias when the two become decoupled (Michaletz et al., 2015). The impact of these simplifications may be minor but should be evaluated.

A further simplification is that investment in electron transport capacity (expressed by $J_{max}$) and investments in the carboxylation capacity (expressed by $V_{cmax}$) are coordinated so that for conditions with which the model is forced (here, monthly means of daily averages), photosynthesis operates at the co-limitation point of the light- and Rubisco-limited assimilation rates and an
effective linear relationship between absorbed light and mean assimilation emerges. This assumption follows from the *coordination hypothesis* (Chen et al., 1993; Haxeltine and Prentice, 1996), which itself can be understood as an optimality principle (Haxeltine and Prentice, 1996), and is well supported by observations (Maire et al., 2012). However, this coordination is contingent on the time scale at which photosynthetic acclimation occurs, which is not known precisely (Smith and Dukes, 2013; Way and Yamori, 2014). By simulating $\chi$ usingh monthly mean meteorological variables, we assume a monthly time scale of
acclimation. This is probably a conservative estimate (Smith and Dukes, 2017; Veres and Williams, 1984). Considering the concave relationship of assimilation rates and absorbed light that follows from the FvCB model for a given $J_{max}$, linearly scaling a given monthly LUE term with daily varying absorbed light levels should lead to an overestimation of assimilation rates at high light levels. This overestimation should disappear as the time scale over which light levels are averaged is increased. However, our results do not confirm these expectations. The fact that the model did not exhibit a systematic error in simulating
GPP variations when applied at the daily time scale suggests that the day-to-day variability in light levels is relatively small compared to the within-day variability.



## 6  Conclusions

The P-model provides a simple, parameter-sparse but powerful method to predict photosynthetic capacity and light use efficiency across a wide range of climatic conditions and vegetation types. It provides a basis for a terrestrial light use efficiency model driven by remotely sensed vegetation greenness. Using optimality principles for the formulation of the P-model reduces its dependence on uncertain or vegetation type-specific parameters and enables robust predictions of GPP and its variations through the seasons, between years, and across space. Further work is required to develop a distinct treatment of $C_4$ vegetation for global applications and additional evaluations are needed to examine the P-model's sensitivity to increasing $CO_2$. We have shown that accounting for the effects of low soil moisture and the reduction in the quantum yield efficiency under low temperatures improves model performance. There is potential to include below-ground water limitation effects in the mechanistic optimality framework of the P-model.

*Code and data availability.*  The P-model is implemented as an R package (*rpmodel*) and available through https://stineb.github.io/rpmodel/. The R package will be made available through CRAN. Code for all evaluations presented here is available through https://github.com/stineb/eval_pmodel. Model outputs are available on *Zenodo* http://doi.org/10.5281/zenodo.3247930.

## Appendix A:  Site information

Table A1 provides meata information and references for each site from the FLUXNET2015 Tier 1 dataset, used for model calibration and evaluation in the present study.

## Appendix B:  Temperature and pressure dependence of photosynthesis parameters

### B1  Photorespiratory Compensation Point $\Gamma^*$

The temperature and pressure-dependent photorespiratory compensation point in absence of dark respiration $\Gamma^*(T,p)$ is calculated from its value at standard temperature ($T_0 = 25°C$) and atmospheric pressure ($p_0 =$101325 Pa), referred to as $\Gamma^*_{25,p_0}$. It is modified by temperature following an Arrhenius-type temperature response function $f_{\text{Arrh}}(T_K, \Delta H_{\Gamma*})$ with activation energy $\Delta H_{\Gamma*}$, and is corrected for atmospheric pressure $p(z)$ at elevation $z$.

$$\Gamma^*(T_K, z) = \Gamma^*_{25,p_0} \; f_{\text{Arrh}}(T_K, \Delta H_{\Gamma*}) \, \frac{p(z)}{p_0} \tag{B1}$$

Values of $\Delta H_{\Gamma*}$ and $\Gamma^*_{25,p_0}$ are taken from Bernacchi et al. (2001). The latter is converted to Pa and standardised to $p_0$ simply by multiplication with $p_0$ ($\Gamma^*_{25,p_0} = 42.75\ \mu\text{mol mol}^{-1} \cdot 10^{-6} \cdot 101325\ \text{Pa} = 4.332\ \text{Pa}$). $\Delta H_{\Gamma*}$ is 37830 J mol$^{-1}$. All parameter values are summarised in Tab. A7. The function $p(z)$ is defined in Sec B4. Note that $T_K$ indicates that the respective temperature value is given in Kelvin and $T_{K,0} = 298.15$ K.





To correct for effects by temperature following the Arrhenius Equation with its form $x(T_K) = \exp(c - \Delta H_a/(T_K R))$, the temperature-correction function $f_{\text{Arrh}}(T_K, \Delta H_a)$, used in Eq. B1 and further equations below, is given by:

$$f_{\text{Arrh}}(T_K) = x(T_K)/x(T_{K,0}) = \exp\left(\frac{\Delta H(T_K - T_{K,0})}{T_{K,0}\, R\, T_K}\right) \tag{B2}$$

where $\Delta H$ is the respective activation energy (e.g., $\Delta H_{\Gamma*}$ in Eq. B1), and $R$ is the universal gas constant (8.3145 J mol$^{-1}$ K$^{-1}$).

## B2 Deriving $\Gamma^*$

The temperature and pressure dependency of $\Gamma^*$ follows from the temperature dependencies of $K_c$, $K_o$, $V_{c,\max}$, and $V_{o,\max}$ and the pressure dependency of $pO_2(p)$:

$$\Gamma^*(T_K, p) = \frac{pO_2(p)\, K_c(T_K)\, V_{\text{omax}}(T_K)}{2\, K_o(T_K)\, V_{\text{cmax}}(T_K)} \tag{B3}$$

$pO_2(p)$ is the partial pressure of atmospheric oxygen (Pa) and scales linearly with $p(z)$. $K_c$ is the Michaelis-Menten constant for carboxylation (Pa); $K_o$ is the Michaelis-Menten constant for oxygenation (Pa); $V_{\text{cmax}}$ is maximum rate of carboxylation ($\mu$mol m$^{-2}$ s$^{-1}$); and $V_{\text{omax}}$ is the maximum rate of oxygenation ($\mu$mol m$^{-2}$ s$^{-1}$). The temperature-dependency equations for these four terms are given in Table 1 of Bernacchi et al. (2001) with respective scaling constants $c$ and activation energies $\Delta H_a$ as :

$$K_c(T_K) = \exp(38.05 - 79.43/(T_K R)) \tag{B4a}$$

$$K_o(T_K) = 1000 \cdot \exp(20.30 - 36.38/(T_K R)) \tag{B4b}$$

$$V_{\text{o,max}}(T_K) = \exp(22.98 - 60.11/(T_K R)) \tag{B4c}$$

$$V_{\text{c,max}}(T_K) = \exp(26.35 - 65.33/(T_K R)) \tag{B4d}$$

By substituting the temperature-dependency equations for each term in Eq. B3 and rearranging terms, $\Gamma^*$ can be written as

$$\Gamma^*(T_K, z) = pO_2(z)\, \exp(6.779 - 37.83/(T_K R)) \,. \tag{B5}$$

With $pO_2(p)$ at standard atmospheric pressure (101325 Pa) taken to be 21000 Pa, and assuming a constant mixing ratio across the troposphere, its pressure dependence can be expressed as

$$pO_2(p) = 0.2095 \cdot p(z) \tag{B6}$$

hence

$$\Gamma^*(T_K, p) = p(z)\, \exp(5.205 - 37.83/(T_K R)) \tag{B7}$$

We can use this to calculate $\Gamma^*$ at standard temperature ($T_K = 298.15$ K) and pressure ($p(z) = 101325$ Pa) as $\Gamma^*_{25, p_0} = 4.332$ Pa.




Note that to convert Eq. B5 to the form corresponding to the one given by Bernacchi et al. (2001), the partial pressure of oxygen ($pO_2$) has to be assumed at standard conditions. $pO_2$ is approximately 21000 Pa and with the standard atmospheric pressure of 101325 Pa, $pO_2$ can be converted from Pascals to parts-per-million (ppm) as $21000/101325 \times 10^6 = 207254$ ppm $= \exp(12.24)$ ppm. This can be combined with the exponent in Eq. B5 to $\exp(12.24) \cdot \exp(6.779) = \exp(19.02)$. This

corresponds to the parameter values determining the temperature dependence of $\Gamma^*$ given by Bernacchi et al. (2001) as $\Gamma^* = \exp(19.02 - 37.83/(T_K R))$.

**B3   Michaelis-Menten Coefficient of Photosynthesis**

The effective Michaelis-Menten coefficient $K$ (Pa) of Rubisco-limited photosynthesis (Eq. 6) is determined by the Michaelis-Menten constants for the carboxylation and oxygenation reactions (Farquhar et al., 1980):

$$K(T_K, p) = K_c(T_K) \left(1 + \frac{pO_2(p)}{K_o(T_K)}\right) , \tag{B8}$$

where $K_c$ is the Michaelis-Menten constant for $CO_2$ (Pa), $K_o$ is the Michaelis-Menten constant for the carboxylation and oxygenation reaction, respectively, and $pO_2$ is the partial pressure of oxygen (Pa). $K_c$ and $K_o$ follow a temperature dependence, given by the Arrhenius equation analogously to the temperature dependence of $\Gamma^*$ (Eq. B1):

$$K_c(T_K) = K_{c25} \, f_{\text{Arrh}}(T_K, \Delta H_{Kc}) \tag{B9a}$$

$$K_o(T_K) = K_{o25} \, f_{\text{Arrh}}(T_K, \Delta H_{Ko}) \tag{B9b}$$

Values $\Delta H_{Kc} = 79430$ J mol$^{-1}$, $\Delta H_{Ko} = 36380$ J mol$^{-1}$, $K_{c25} = 39.97$ Pa, and $K_{o25} = 27480$ Pa are taken from Bernacchi et al. (2001) and (see also Tab. A7). The latter two have been converted from $\mu$mol mol$^{-1}$ in Bernacchi et al. (2001) to units of Pa by multiplication with the standard atmosphere (101325 Pa). Note that $K_{c25}$ and $K_{o25}$ are rate constants and are independent of atmospheric pressure. Pressure-dependence of $K$ is solely in $pO_2(p)$ (see Eq. B6).

**B4   Atmospheric pressure**

The elevation-dependence of atmospheric pressure is computed by assuming a linear decrease in temperature with elevation and a mean adiabatic lapse rate (Berberan-Santos et al., 1997):

$$p(z) = p_0 \left(1 - \frac{Lz}{T_{K,0}}\right)^{gM_a(RL)^{-1}} , \tag{B10}$$

where $z$ is the elevation above mean sea level (m), $g$ is the gravitational constant (9.80665 m s$^{-2}$), $p_0$ is the standard atmospheric
pressure at 0 m a.s.l. (101325 Pa), $L$ is the mean adiabatic lapse rate (0.0065 K m$^{-2}$), $M_a$ is the molecular weight for dry air (0.028963 kg mol$^{-1}$), and $R$ is the universal gas constant (8.3145 J mol$^{-1}$ K$^{-1}$). All parameter values that are held fixed in the model (not calibrated) are summarised in Tab. A7.





**Appendix C: Corollary of the $\chi$ prediction**

**C1  Stomatal conductance**

Stomatal conductance $g_s$ (mol C Pa$^{-1}$) follows from the prediction of $\chi$ given by Eq. 8 and $g_s = A/(c_a\,(1-\chi))$ (from Eq. 5).
Stomatal contuctance can thus be written as

$$g_s = \left(1 + \frac{\xi}{\sqrt{D}}\right)\frac{A}{c_a - \Gamma^*}\;. \tag{C1}$$

This has a similar form as the solution for $g_s$ derived from a different optimality principle by Medlyn et al. (2011) (their Eq.
11). Differences are that an additional term $g_0$ is missing here and that $\Gamma^*$ does not appear in Medlyn et al. (2011). The theory
presented by Prentice et al. (2014) provides a theoretical interpretation for the parameter $g_1$ in Medlyn et al. (2011): It is given
by $\xi$ (Eq. 9) and can thus be predicted from the environment. However, it is notable that the underlying optimality criterion used
by Medlyn et al. (2011), as proposed by Cowan and Farquhar (1977), is one that maintains a constant marginal water cost of
carbon gain $\lambda = \partial E/\partial A$. It thus describes an instantaneous $g_s$ adjustment, e.g., to diurnal variations in $D$ and has been adopted
into DVMs and ESMs for respective predictions (with a given $V_{\mathrm{cmax}}$). In contrast, the theory presented here and underlying the
P-model predicts $\chi$ which is jointly controlled by $g_s$ and $V_{\mathrm{cmax}}$. In other words, it predicts a $g_s$ that is coordinated with $V_{\mathrm{cmax}}$
and thus acclimates at a similar time scale (which is on the order of days to weeks). This $\chi$ can be understood as a "set-point"
for an average $\chi$ with actual $\chi$ varying around it at a daily to sub-daily time scale.

**C2  Intrinsic water use efficiency**

The intrinsic water use efficiency (iWUE, in Pa) has been defined as the ratio of assimilation over stomatal conductance (to
water) (Beer et al., 2009) as iWUE $= A/(1.6g_s)$. The factor 1.6 accounts for the difference in diffusivity between $CO_2$ and
$H_2O$. Using Fick's Law (Eq. 5), this is simply

$$\text{iWUE} = \frac{c_a(1-\chi)}{1.6}\;, \tag{C2}$$

or, using the prediction of optimal $\chi$ given by Eq. 8, this can be expressed as

$$\text{iWUE} = \frac{1}{1.6\left(1 + \frac{\xi}{\sqrt{D}}\right)}\,(c_a - \Gamma^*) \tag{C3}$$

**C3  Maximum carboxylation capacity**

$V_{\mathrm{cmax}}$ With $A_J = A_C$, $V_{\mathrm{cmax}}$ can directly be derived as

$$V_{\mathrm{cmax}} = \varphi_0\,I_{\mathrm{abs}}\,\frac{c_i + K}{c_i + 2\Gamma^*} = \varphi_0\,I_{\mathrm{abs}}\,\frac{m}{m_C}\;, \tag{C4}$$

$c_i$ is given by $c_a\chi$. The second part of the equation follows from the definitions of $m$ (Eq. 11) and $m_C$ (Eq. 7). Normalising
$V_{\mathrm{cmax}}$ to standard temperature (25°C) following a modified Arrhenius function based on Kattge and Knorr (2007) gives $V_{\mathrm{cmax25}}$





as

$$V_{\mathrm{cmax25}} = V_{\mathrm{cmax}}/f_V(T_K, T_{K,0}) \tag{C5}$$

$$f_V(T_K, T_{K,0}) = f_{\mathrm{Arrh}}(T_K, \Delta H_V) \cdot \frac{1 + \exp((T_{K,0}\Delta S - H_d)/(T_{K,0}R))}{1 + \exp((T_K\Delta S - H_d)/(T_K R))} \tag{C6}$$

with $H_V$ being the activation energy (71513 J mol$^{-1}$), $H_d$ is the deactivation energy (200000 J mol$^{-1}$), and $\Delta S$ is an entropy term (J mol$^{-1}$ K$^{-1}$) calculated using a linear relationship with $T$ from Kattge and Knorr (2007), with a slope of $b_S = 1.07$ J

mol$^{-1}$ K$^{-2}$ and intercept of $a_S = 668.39$ J mol$^{-1}$ K$^{-1}$:

$$\Delta S = a_S - b_S T \tag{C7}$$

Note that $T$ is in units of °C in above equation. Equation C6 describes the *instantaneous* response to temperature and is not the same as the optimality-driven *acclimation* to temperature predicted by the P-model.

### C4   Dark respiration $R_{\mathrm{d}}$

Dark respiration at standard temperature $R_{\mathrm{d25}}$ is calculated as being proportional to $V_{\mathrm{cmax25}}$:

$$R_{\mathrm{d25}} = b_0 \, V_{\mathrm{cmax25}} \tag{C8}$$

where $b_0 = 0.015$ (Atkin et al., 2015). Dark respiration follows a slightly different instantaneous temperature sensitivity than $V_{\mathrm{cmax}}$ following Heskel et al. (2016):

$$R_{\mathrm{d}} = R_{\mathrm{d25}} \, f_R \tag{C9}$$

$$f_R = \exp\left(0.1012(T_{K,0} - T_K) - 0.0005(T_{K,0}^2 - T_K^2)\right) \tag{C10}$$

By combining Eqs. C6, C8, and C9, $R_d$ at growth temperature $T$ can directly be calculated from $V_{\mathrm{cmax}}$ as

$$R_d = b_0 \frac{f_R}{f_V} \, V_{\mathrm{cmax}} \tag{C11}$$

### Appendix D: Soil water holding capacity

The soil water balance is solved following the SPLASH model. Precipitation in the form of rain ($P_{\mathrm{rain}}$) and snow ($P_{\mathrm{snow}}$)

are taken from WATCH-WFDEI (Weedon et al., 2014) and are summed and converted from kg m$^{-2}$ s$^{-1}$ to mm d$^{-1}$ by multiplication with $(60 \cdot 60 \cdot 24)$ s d$^{-1}$. To obtain the total soil water holding capacity (WHC, in mm), we use soil depth-to-bedrock and texture data from SoilGrids (Hengl et al., 2014), extracted around the FLUXNET sites. We assumed that the plant-available WHC is determined by the WHC down to a maximum depth of 2 m and limited by the depth-to-bedrock. The water holding capacity ($w_{\mathrm{WHC}}$, in mm) was defined as the difference in volumetric soil water storage at field capacity ($W_{\mathrm{FC}}$, in

m$^3$ m$^{-3}$) and the permanent wilting point ($W_{\mathrm{PWP}}$, in m$^3$ m$^{-3}$):

$$\theta_{\mathrm{WHC}} = (W_{\mathrm{FC}} - W_{\mathrm{PWP}}) \, (1 - f_{\mathrm{gravel}}) \cdot \min(z_{\mathrm{bedrock}}, z_{\mathrm{max}}) \tag{D1}$$





$f_{\text{grave}}$ is the gravel fraction, $z_{\text{bedrock}}$ is the depth to bedrock (in m), and $z_{\text{max}}$ is 2 m. The volumetric soil water storage at field capacity and wilting point were obtained from texture and organic matter content data through pedotransfer functions, as described by Saxton and Rawls (2006). The volumetric soil water storage (m$^3$ m$^{-3}$) at field capacity is calculated as:

$$W_{\text{FC}} = k_{\text{FC}} + \left(1.283 \cdot k_{\text{FC}}^2 - 0.374 \cdot k_{\text{FC}} - 0.015\right), \tag{D2}$$

where

$$k_{\text{FC}} = -0.251 \cdot f_{\text{sand}} + 0.195 \cdot f_{\text{clay}} + 0.011 \cdot f_{\text{OM}} \tag{D3}$$
$$+ 0.006 \cdot (f_{\text{sand}} f_{\text{OM}}) \tag{D4}$$
$$- 0.027 \cdot (f_{\text{clay}} f_{\text{OM}}) \tag{D5}$$
$$+ 0.452 \cdot (f_{\text{sand}} f_{\text{clay}}) \tag{D6}$$
$$+ 0.299 \tag{D7}$$

$f_{\text{sand}}$, $f_{\text{clay}}$, $f_{\text{OM}}$ are the sand, clay and organic matter contents in percent by weight. The volumetric soil water storage (m$^3$ m$^{-3}$) at the permanent wilting point is calculated as:

$$W_{\text{PWP}} = k_{\text{PWP}} + \left(0.14 \cdot k_{\text{PWP}} - 0.02\right), \tag{D8}$$

where

$$k_{\text{PWP}} = -0.024 \cdot f_{\text{sand}} + 0.487 \cdot f_{\text{clay}} + 0.006 \cdot f_{\text{OM}} \tag{D9}$$
$$+ 0.005 \cdot (f_{\text{sand}} f_{\text{OM}}) \tag{D10}$$
$$- 0.013 \cdot (f_{\text{clay}} f_{\text{OM}}) \tag{D11}$$
$$+ 0.068 \cdot (f_{\text{sand}} f_{\text{clay}}) \tag{D12}$$
$$+ 0.031 \tag{D13}$$

## Appendix E: Extended theory

### E1 Deriving $\chi$

Using Eqs. 4 and 5, the term on the left-hand side of Eq. 3 can thus be written as

$$\frac{\partial (E/A)}{\partial \chi} = \frac{1.6\,D}{c_a\,(1-\chi)^2}\,. \tag{E1}$$

Using Equation 6 and the simplification $\Gamma^* = 0$, the derivative term on the right-hand-side of Eq.3 can be written as

$$\frac{\partial (V_{\text{cmax}}/A)}{\partial \chi} = -\frac{K}{c_a\,\chi^2}\,. \tag{E2}$$





Eq. 3 can thus be written as

$$a\,\frac{1.6\,D}{c_a\,(1-\chi)^2} = b\,\frac{K}{c_a\,\chi^2} \tag{E3}$$

and solved for $\chi$:

$$\chi = \frac{\xi}{\xi + \sqrt{D}} \tag{E4}$$

$$\xi = \sqrt{\frac{\beta K}{1.6\eta^*}} \tag{E5}$$

Where $b/a = \beta/\eta^*$. The exact solution, without the simplification $\Gamma^* = 0$, and following analogous steps, is

$$\chi = \frac{\Gamma^*}{c_a} + \left(1 - \frac{\Gamma^*}{c_a}\right)\frac{\xi}{\xi + \sqrt{D}} \tag{E6}$$

$$\xi = \sqrt{\frac{b(K + \Gamma^*)}{1.6\,a}} \tag{E7}$$

This can also be written as

$$c_i = \frac{\Gamma^*\sqrt{D} + \xi\,c_a}{\xi + \sqrt{D}}\;. \tag{E8}$$

### E2   Deriving the $J_{\mathrm{max}}$ limitation factor

By taking the derivative of $A_J$ with respect to $J_{\mathrm{max}}$, Eq. 14 can be expressed as

$$c = \frac{m(\varphi_0 I_{\mathrm{abs}})^3}{4\sqrt{\left[(\varphi_0 I_{\mathrm{abs}})^2 + (\frac{J_{\mathrm{max}}}{4})^2\right]^3}} \tag{E9}$$

This can be re-arranged to

$$\left(\frac{4c}{m}\right)^{2/3} = \frac{1}{1 + \left(\frac{J_{\mathrm{max}}}{4\varphi_0 I_{\mathrm{abs}}}\right)^2} \tag{E10}$$

For simplification, we can substitute

$$k = \frac{4\varphi_0 I_{\mathrm{abs}}}{J_{\mathrm{abs}}} \tag{E11}$$

and

$$u = \left(\frac{4c}{m}\right)^{2/3} \tag{E12}$$

With this, we can write

$$\frac{1}{1 + k^{-2}} = u\;. \tag{E13}$$

This can be re-arranged to

$$(1 - u)^{1/2} = \frac{1}{\sqrt{1 + k^2}} \tag{E14}$$

The right-hand term now corresponds to the $J_{\mathrm{max}}$ limitation factor $L$ in Eq. 13, and we get Eq. 15.





### E3   An alternative method for introducing the $J_{\text{max}}$ limitation

Sect. 2.2 introduced the effect of a finite $J_{\text{max}}$ leading to a saturating relationship between absorbed light and the light-limited assimilation rate $A_J$. An alternative method was presented by Smith et al. (2019) and is implemented in *rpmodel* as an optional method (argument `method_jmaxlim = "smith19"`). Following their approach, the light-limited assimilation rate is described as

$$A_J = \left(\frac{J}{4}\right) m \,. \tag{E15}$$

$m$ is the $CO_2$ limitation factor (Eq. 11), and $J$ is a saturating function of absorbed light, approaching $J_{\text{max}}$ for high light levels, following Farquhar et al. (1980):

$$\theta J^2 - (\varphi_0 I_{\text{abs}} + J_{\text{max}}) J + \varphi_0 I_{\text{abs}} J_{\text{max}} = 0 \,. \tag{E16}$$

$\theta$ is a unitless parameter determining the curvature of the response of $J$ to $I_{\text{abs}}$, here taken as 0.85, based on Smith et al. (2019) and references therein. Eq. E16 can be substituted into Eq. E15 to yield

$$A_J = \left(\frac{m}{4}\right) \frac{\varphi_0 I_{\text{abs}} + J_{\text{max}} \pm \sqrt{(\varphi_0 I_{\text{abs}} + J_{\text{max}})^2 - 4\theta\varphi_0 I_{\text{abs}} J_{\text{max}}}}{2\theta} \,, \tag{E17}$$

from which the smaller root is used to derive $A_J$. Similar as in the method used by Wang et al. (2017a) and outlined in Sect. 2.2, a proportionality between $A_J$ and $J_{\text{max}}$ is assumed ($\partial A / \partial J_{\text{max}} = c$; Eq. 14). Taking the derivative of Eq. E17 with respect to $J_{\text{max}}$ and setting equal to $c$ leads to

$$J_{\text{max}} = \varphi_0 \, I_{\text{abs}} \, \omega \tag{E18}$$

with

$$\omega = -(1 - 2\theta) + \sqrt{(1 - \theta)\left(\frac{1}{\frac{4c}{m}\left(1 - \theta\frac{4c}{m}\right)} - 4\theta\right)} \,. \tag{E19}$$

Using this, $A_J$ can be written analogously to Eq. 16, but with

$$m' = m \, \frac{\omega^*}{8\theta} \,, \tag{E20}$$

and

$$\omega^* = 1 + \omega - \sqrt{(1 + \omega)^2 - 4\theta\omega} \,. \tag{E21}$$

The cost parameter $c$ was assumed to be non-varying. Under standard conditions of 25 °C, 101325 Pa atmospheric pressure, 1000 Pa vapor pressure deficit, and 360 ppm $CO_2$, at which the ratio of $J_{\text{max}}$ to $V_{\text{cmax}}$ was assumed to be 2.07 (Smith and Dukes, 2017), $c$ was derived as 0.053 (Smith et al., 2019).

Using the definition of $V_{\text{cmax}}$ from Eq. C4, $m$ can be replaced by $m'$ from Eq. E20 to calculate an "intermediate rate of $V_{\text{cmax}}$" (Smith et al., 2019) as

$$V_{\text{cmax}} = \varphi_0 \, I_{\text{abs}} \, \frac{m'}{m_C} \tag{E22}$$





## Appendix F: The `rpmodel()` function of the *rpmodel* R package

The *rpmodel* R package provides an implementation of the P-model as described here. The main function is `rpmodel()`
which returns a list of variables that are mutually consistent within the theory of the P-model (Sect. 2) and based on calculations
defined in this paper. References for the returned list of variables are given in Tab. A8





**Table A1.** Sites used for evaluation. Lon. is longitude, negative values indicate west longitude; Lat. is latitude, positive values indicate north latitude; Veg. is vegetation type: deciduous broadleaf forest (DBF); evergreen broadleaf forest (EBF); evergreen needleleaf forest (ENF); grassland (GRA); mixed deciduous and evergreen needleleaf forest (MF); savanna ecosystem (SAV); shrub ecosystem (SHR); wetland (WET).

| Site | Lon. | Lat. | Period | Veg. | Clim. | N | Calib. | Reference |
|------|------|------|--------|------|-------|---|--------|-----------|
| AR-SLu | -66.46 | -33.46 | 2009-2011 | MF | Bwk | 446 | | Ulke et al. (2015) |
| AR-Vir | -56.19 | -28.24 | 2009-2012 | ENF | Csb | 749 | Y | Posse et al. (2016) |
| AT-Neu | 11.32 | 47.12 | 2002-2012 | GRA | Dfc | 3243 | | Wohlfahrt et al. (2008) |
| AU-Ade | 131.12 | -13.08 | 2007-2009 | WSA | Aw | 532 | Y | Beringer et al. (2011a) |
| AU-ASM | 133.25 | -22.28 | 2010-2013 | ENF | BSh | 1045 | Y | Cleverly et al. (2013) |
| AU-Cpr | 140.59 | -34.00 | 2010-2014 | SAV | BSk | 1370 | | Meyer et al. (2015) |
| AU-Cum | 150.72 | -33.61 | 2012-2014 | EBF | Cfa | 744 | | Beringer et al. (2016a) |
| AU-DaP | 131.32 | -14.06 | 2007-2013 | GRA | Aw | 1402 | Y | Beringer et al. (2011b) |
| AU-DaS | 131.39 | -14.16 | 2008-2014 | SAV | Aw | 2265 | Y | Hutley et al. (2011) |
| AU-Dry | 132.37 | -15.26 | 2008-2014 | SAV | Aw | 1598 | Y | Cernusak et al. (2011) |
| AU-Emr | 148.47 | -23.86 | 2011-2013 | GRA | Bwk | 755 | | Schroder et al. (2014) |
| AU-Fog | 131.31 | -12.55 | 2006-2008 | WET | Aw | 878 | Y | Beringer et al. (2013) |
| AU-Gin | 115.71 | -31.38 | 2011-2014 | WSA | Csa | 942 | Y | Beringer et al. (2016b) |
| AU-GWW | 120.65 | -30.19 | 2013-2014 | SAV | Bwk | 663 | | Prober et al. (2012) |
| AU-Lox | 140.66 | -34.47 | 2008-2009 | DBF | Bsh | 273 | | Stevens et al. (2011) |
| AU-RDF | 132.48 | -14.56 | 2011-2013 | WSA | Bwh | 431 | | Bristow et al. (2016) |
| AU-Rig | 145.58 | -36.65 | 2011-2014 | GRA | Cfb | 1130 | | Beringer et al. (2016c) |
| AU-Rob | 145.63 | -17.12 | 2014-2014 | EBF | Csb | 337 | | Beringer et al. (2016d) |
| AU-Stp | 133.35 | -17.15 | 2008-2014 | GRA | BSh | 1318 | Y | Beringer et al. (2011c) |
| AU-TTE | 133.64 | -22.29 | 2012-2013 | OSH | BWh | 94 | | Cleverly et al. (2016) |
| AU-Tum | 148.15 | -35.66 | 2001-2014 | EBF | Cfb | 4335 | | Leuning et al. (2005) |
| AU-Wac | 145.19 | -37.43 | 2005-2008 | EBF | Cfb | 979 | | Kilinc et al. (2013) |
| AU-Whr | 145.03 | -36.67 | 2011-2014 | EBF | Cfb | 1065 | Y | McHugh et al. (2017) |
| AU-Wom | 144.09 | -37.42 | 2010-2012 | EBF | Cfb | 934 | Y | Hinko-Najera et al. (2017) |
| AU-Ync | 146.29 | -34.99 | 2012-2014 | GRA | BSk | 392 | | Yee et al. (2015) |
| BE-Bra | 4.52 | 51.31 | 1996-2014 | MF | Cfb | 4208 | Y | Carrara et al. (2004) |
| BE-Vie | 6.00 | 50.31 | 1996-2014 | MF | Cfb | 4733 | Y | Aubinet et al. (2001) |
| BR-Sa3 | -54.97 | -3.02 | 2000-2004 | EBF | Am | 1206 | | Wick et al. (2005) |
| CA-Man | -98.48 | 55.88 | 1994-2008 | ENF | Dfc | 1411 | | Dunn et al. (2007) |
| CA-NS1 | -98.48 | 55.88 | 2001-2005 | ENF | Dfc | 771 | | Goulden et al. (2006a) |
| CA-NS2 | -98.52 | 55.91 | 2001-2005 | ENF | Dfc | 873 | | Goulden et al. (2006b) |



**Table A2.** Continued from Table A1

| Site | Lon. | Lat. | Period | Veg. | Clim. | N | Calib. | Reference |
|------|------|------|--------|------|-------|---|--------|-----------|
| CA-NS3 | -98.38 | 55.91 | 2001-2005 | ENF | Dfc | 1069 | | Goulden et al. (2006c) |
| CA-NS4 | -98.38 | 55.91 | 2002-2005 | ENF | Dfc | 610 | | Goulden et al. (2006d) |
| CA-NS5 | -98.48 | 55.86 | 2001-2005 | ENF | Dfc | 912 | | Goulden et al. (2006e) |
| CA-NS6 | -98.96 | 55.92 | 2001-2005 | OSH | Dfc | 913 | | Goulden et al. (2006f) |
| CA-NS7 | -99.95 | 56.64 | 2002-2005 | OSH | Dfc | 709 | | Goulden et al. (2006g) |
| CA-Qfo | -74.34 | 49.69 | 2003-2010 | ENF | Dfc | 1812 | | Bergeron et al. (2007) |
| CA-SF1 | -105.82 | 54.48 | 2003-2006 | ENF | Dfc | 525 | | Mkhabela et al. (2009a) |
| CA-SF2 | -105.88 | 54.25 | 2001-2005 | ENF | Dfc | 675 | | Mkhabela et al. (2009b) |
| CA-SF3 | -106.01 | 54.09 | 2001-2006 | OSH | Dfc | 651 | | Mkhabela et al. (2009c) |
| CH-Cha | 8.41 | 47.21 | 2005-2014 | GRA | Cfb | 2885 | | Merbold et al. (2014) |
| CH-Dav | 9.86 | 46.82 | 1997-2014 | ENF | ET | 4444 | | Zielis et al. (2014) |
| CH-Fru | 8.54 | 47.12 | 2005-2014 | GRA | Cfb | 2566 | Y | Imer et al. (2013) |
| CH-Lae | 8.37 | 47.48 | 2004-2014 | MF | Cfb | 3204 | Y | Etzold et al. (2011) |
| CH-Oe1 | 7.73 | 47.29 | 2002-2008 | GRA | Cfb | 2104 | Y | Ammann et al. (2009) |
| CN-Cha | 128.10 | 42.40 | 2003-2005 | MF | Dwb | 982 | | Guan et al. (2006) |
| CN-Cng | 123.51 | 44.59 | 2007-2010 | GRA | Bsh | 1113 | Y | ? |
| CN-Dan | 91.07 | 30.50 | 2004-2005 | GRA | ET | 647 | | Shi et al. (2006) |
| CN-Din | 112.54 | 23.17 | 2003-2005 | EBF | Cfa | 917 | | Yan et al. (2013) |
| CN-Du2 | 116.28 | 42.05 | 2006-2008 | GRA | Dwb | 616 | | Chen et al. (2009) |
| CN-Ha2 | 101.33 | 37.61 | 2003-2005 | WET | ET | 1030 | | ? |
| CN-HaM | 101.18 | 37.37 | 2002-2004 | GRA | | 688 | | Kato et al. (2006) |
| CN-Qia | 115.06 | 26.74 | 2003-2005 | ENF | Cfa | 992 | Y | Wen et al. (2010) |
| CN-Sw2 | 111.90 | 41.79 | 2010-2012 | GRA | Bsh | 237 | | Shao et al. (2017) |
| CZ-BK1 | 18.54 | 49.50 | 2004-2008 | ENF | Dfb | 1100 | | Acosta et al. (2013) |
| CZ-BK2 | 18.54 | 49.49 | 2004-2006 | GRA | Dfb | 161 | | ? |
| CZ-wet | 14.77 | 49.02 | 2006-2014 | WET | Cfb | 2605 | Y | Dušek et al. (2012) |
| DE-Gri | 13.51 | 50.95 | 2004-2014 | GRA | Cfb | 3387 | Y | Prescher et al. (2010) |
| DE-Hai | 10.45 | 51.08 | 2000-2012 | DBF | Cfb | 3435 | Y | Knohl et al. (2003) |
| DE-Lkb | 13.30 | 49.10 | 2009-2013 | ENF | Cfb | 1001 | | Lindauer et al. (2014) |
| DE-Obe | 13.72 | 50.78 | 2008-2014 | ENF | Cfb | 2043 | Y | ? |
| DE-RuR | 6.30 | 50.62 | 2011-2014 | GRA | Cfb | 1195 | Y | Post et al. (2015) |





**Table A3.** Continued from Table A1

| Site | Lon. | Lat. | Period | Veg. | Clim. | N | Calib. | Reference |
|------|------|------|--------|------|-------|---|--------|-----------|
| DE-SfN | 11.33 | 47.81 | 2012-2014 | WET | Cfb | 750 | | Hommeltenberg et al. (2014) |
| DE-Spw | 14.03 | 51.89 | 2010-2014 | WET | Cfb | 1339 | Y | ? |
| DE-Tha | 13.57 | 50.96 | 1996-2014 | ENF | Cfb | 4887 | Y | Grünwald and Bernhofer (2007) |
| DK-NuF | -51.39 | 64.13 | 2008-2014 | WET | ET | 882 | Y | Westergaard-Nielsen et al. (2013) |
| DK-Sor | 11.64 | 55.49 | 1996-2014 | DBF | Cfb | 4483 | Y | Pilegaard et al. (2011) |
| DK-ZaF | -20.55 | 74.48 | 2008-2011 | WET | ET | 381 | | Stiegler et al. (2016) |
| DK-ZaH | -20.55 | 74.47 | 2000-2014 | GRA | ET | 1696 | | Lund et al. (2012) |
| ES-LgS | -2.97 | 37.10 | 2007-2009 | OSH | Csa | 794 | | Reverter et al. (2010) |
| ES-Ln2 | -3.48 | 36.97 | 2009-2009 | OSH | Csa | 69 | | Serrano-Ortiz et al. (2011) |
| FI-Hyy | 24.30 | 61.85 | 1996-2014 | ENF | Dfc | 4222 | Y | Suni et al. (2003) |
| FI-Lom | 24.21 | 68.00 | 2007-2009 | WET | Dfc | 575 | | Aurela et al. (2015) |
| FI-Sod | 26.64 | 67.36 | 2001-2014 | ENF | Dfc | 2816 | Y | Thum et al. (2007) |
| FR-Fon | 2.78 | 48.48 | 2005-2014 | DBF | Cfb | 2827 | Y | Delpierre et al. (2015) |
| FR-LBr | -0.77 | 44.72 | 1996-2008 | ENF | Cfb | 2800 | Y | Berbigier et al. (2001) |
| FR-Pue | 3.60 | 43.74 | 2000-2014 | EBF | Csa | 4723 | Y | Rambal et al. (2004) |
| GF-Guy | -52.92 | 5.28 | 2004-2014 | EBF | Af | 3719 | | Bonal et al. (2008) |
| IT-CA1 | 12.03 | 42.38 | 2011-2014 | DBF | Csa | 1036 | | Sabbatini et al. (2016a) |
| IT-CA3 | 12.02 | 42.38 | 2011-2014 | DBF | Csa | 913 | | Sabbatini et al. (2016b) |
| IT-Col | 13.59 | 41.85 | 1996-2014 | DBF | Cfa | 2822 | Y | Valentini et al. (1996) |
| IT-Cp2 | 12.36 | 41.70 | 2012-2014 | EBF | Csa | 764 | Y | Fares et al. (2014) |
| IT-Cpz | 12.38 | 41.71 | 1997-2009 | EBF | Csa | 2601 | Y | Garbulsky et al. (2008) |
| IT-Isp | 8.63 | 45.81 | 2013-2014 | DBF | Cfb | 588 | Y | Ferréa et al. (2012) |
| IT-Lav | 11.28 | 45.96 | 2003-2014 | ENF | Cfb | 3919 | Y | Marcolla et al. (2003) |
| IT-MBo | 11.05 | 46.01 | 2003-2013 | GRA | Dfb | 3236 | Y | Marcolla et al. (2011) |
| IT-Noe | 8.15 | 40.61 | 2004-2014 | CSH | Cwb | 3083 | Y | Papale et al. (2014) |
| IT-PT1 | 9.06 | 45.20 | 2002-2004 | DBF | Cfa | 828 | Y | Migliavacca et al. (2009) |
| IT-Ren | 11.43 | 46.59 | 1998-2013 | ENF | Dfc | 3043 | Y | Montagnani et al. (2009) |
| IT-Ro2 | 11.92 | 42.39 | 2002-2012 | DBF | Csa | 2671 | | Tedeschi et al. (2006) |
| IT-SR2 | 10.29 | 43.73 | 2013-2014 | ENF | Csa | 668 | Y | Hoshika et al. (2017) |
| IT-SRo | 10.28 | 43.73 | 1999-2012 | ENF | Csa | 3791 | Y | Chiesi et al. (2005) |
| IT-Tor | 7.58 | 45.84 | 2008-2014 | GRA | Dfc | 1487 | Y | Galvagno et al. (2013) |





**Table A4.** Continued from Table A1

| Site | Lon. | Lat. | Period | Veg. | Clim. | N | Calib. | Reference |
|------|------|------|--------|------|-------|---|--------|-----------|
| JP-MBF | 142.32 | 44.39 | 2003-2005 | DBF | Dfb | 471 | | Matsumoto et al. (2008a) |
| JP-SMF | 137.08 | 35.26 | 2002-2006 | MF | Cfa | 1288 | Y | Matsumoto et al. (2008b) |
| NL-Hor | 5.07 | 52.24 | 2004-2011 | GRA | Cfb | 2131 | Y | Jacobs et al. (2007) |
| NL-Loo | 5.74 | 52.17 | 1996-2013 | ENF | Cfb | 4507 | Y | Moors (2012) |
| NO-Adv | 15.92 | 78.19 | 2011-2014 | WET | ET | 151 | | ? |
| NO-Blv | 11.83 | 78.92 | 2008-2009 | SNO | ET | 112 | | Lüers et al. (2014) |
| RU-Che | 161.34 | 68.61 | 2002-2005 | WET | Dfc | 313 | | Merbold et al. (2009b) |
| RU-Cok | 147.49 | 70.83 | 2003-2014 | OSH | Dfc | 985 | | van der Molen et al. (2007) |
| RU-Fyo | 32.92 | 56.46 | 1998-2014 | ENF | Dfb | 4042 | Y | Kurbatova et al. (2008) |
| RU-Ha1 | 90.00 | 54.73 | 2002-2004 | GRA | Dfc | 519 | | Marchesini et al. (2007) |
| SD-Dem | 30.48 | 13.28 | 2005-2009 | SAV | BWh | 762 | Y | Ardo et al. (2008) |
| SN-Dhr | -15.43 | 15.40 | 2010-2013 | SAV | BWh | 686 | Y | Tagesson et al. (2014) |
| US-AR1 | -99.42 | 36.43 | 2009-2012 | GRA | Cfa | 1011 | | Raz-Yaseef et al. (2015a) |
| US-AR2 | -99.60 | 36.64 | 2009-2012 | GRA | Cfa | 882 | | Raz-Yaseef et al. (2015b) |
| US-ARb | -98.04 | 35.55 | 2005-2006 | GRA | Cfa | 414 | | Raz-Yaseef et al. (2015c) |
| US-ARc | -98.04 | 35.55 | 2005-2006 | GRA | Cfa | 488 | | Raz-Yaseef et al. (2015d) |
| US-Blo | -120.63 | 38.90 | 1997-2007 | ENF | Csb | 1827 | | Goldstein et al. (2000) |
| US-Cop | -109.39 | 38.09 | 2001-2007 | GRA | BSk | 1067 | | Bowling et al. (2010) |
| US-GBT | -106.24 | 41.37 | 1999-2006 | ENF | Dfc | 541 | | Zeller and Nikolov (2000) |
| US-GLE | -106.24 | 41.37 | 2004-2014 | ENF | Dfb | 2254 | Y | Frank et al. (2014) |
| US-Ha1 | -72.17 | 42.54 | 1991-2012 | DBF | Dfb | 3259 | Y | Urbanski et al. (2007) |
| US-KS2 | -80.67 | 28.61 | 2003-2006 | CSH | Cfa | 1263 | | Powell et al. (2006) |
| US-Los | -89.98 | 46.08 | 2000-2014 | WET | Dfb | 2071 | Y | Sulman et al. (2009) |
| US-Me1 | -121.50 | 44.58 | 2004-2005 | ENF | Csb | 287 | | Irvine et al. (2007) |
| US-Me2 | -121.56 | 44.45 | 2002-2014 | ENF | Csb | 3525 | Y | Irvine et al. (2008) |
| US-Me6 | -121.61 | 44.32 | 2010-2014 | ENF | Csb | 1283 | | Ruehr et al. (2012) |
| US-MMS | -86.41 | 39.32 | 1999-2014 | DBF | Cfa | 3524 | Y | Dragoni et al. (2011) |
| US-Myb | -121.77 | 38.05 | 2010-2014 | WET | Csb | 1153 | | Matthes et al. (2014) |
| US-NR1 | -105.55 | 40.03 | 1998-2014 | ENF | Dfc | 4084 | | Monson et al. (2002) |
| US-PFa | -90.27 | 45.95 | 1995-2014 | MF | Dfb | 3679 | | Desai et al. (2015) |
| US-Prr | -147.49 | 65.12 | 2010-2013 | ENF | Dfc | 546 | | Nakai et al. (2013) |





**Table A5.** Continued from Table A1

| Site | Lon. | Lat. | Period | Veg. | Clim. | N | Calib. | Reference |
|------|------|------|--------|------|-------|---|--------|-----------|
| US-SRG | -110.83 | 31.79 | 2008-2014 | GRA | BSk | 2146 | Y | Scott et al. (2015a) |
| US-SRM | -110.87 | 31.82 | 2004-2014 | WSA | BSk | 3093 | Y | Scott et al. (2009) |
| US-Syv | -89.35 | 46.24 | 2001-2014 | MF | Dfb | 2045 | Y | Desai et al. (2005) |
| US-Ton | -120.97 | 38.43 | 2001-2014 | WSA | Csa | 4321 | Y | Baldocchi et al. (2010) |
| US-Tw1 | -121.65 | 38.11 | 2012-2014 | WET | Csa | 688 | | Oikawa et al. (2017) |
| US-Tw4 | -121.64 | 38.10 | 2013-2014 | WET | Csa | 325 | | Baldocchi (2016) |
| US-UMB | -84.71 | 45.56 | 2000-2014 | DBF | Dfb | 4015 | Y | Gough et al. (2013a) |
| US-UMd | -84.70 | 45.56 | 2007-2014 | DBF | Dfb | 2050 | Y | Gough et al. (2013b) |
| US-Var | -120.95 | 38.41 | 2000-2014 | GRA | Csa | 2981 | Y | Ma et al. (2007) |
| US-WCr | -90.08 | 45.81 | 1999-2014 | DBF | Dfb | 2333 | Y | Cook et al. (2004) |
| US-Whs | -110.05 | 31.74 | 2007-2014 | OSH | BSk | 1561 | | Scott et al. (2015b) |
| US-Wi0 | -91.08 | 46.62 | 2002-2002 | ENF | Dfb | 228 | | Noormets et al. (2007a) |
| US-Wi3 | -91.10 | 46.63 | 2002-2004 | DBF | Dfb | 415 | | Noormets et al. (2007b) |
| US-Wi4 | -91.17 | 46.74 | 2002-2005 | ENF | Dfb | 712 | Y | Noormets et al. (2007c) |
| US-Wi6 | -91.30 | 46.62 | 2002-2003 | OSH | Dfb | 351 | | Noormets et al. (2007d) |
| US-Wi9 | -91.08 | 46.62 | 2004-2005 | ENF | Dfb | 302 | | Noormets et al. (2007e) |
| US-Wkg | -109.94 | 31.74 | 2004-2014 | GRA | BSk | 2676 | | Scott et al. (2010) |
| ZA-Kru | 31.50 | -25.02 | 2000-2010 | SAV | BSh | 2124 | | Archibald et al. (2009) |
| ZM-Mon | 23.25 | -15.44 | 2000-2009 | DBF | Aw | 641 | Y | Merbold et al. (2009a) |





**Table A6.** Fixed parameters. 'SC' stands for 'at standard conditions' (25 °C, 101325 Pa). 'MM coef.' refers to 'Michaelis Menten coefficient'.

| Symbol | Value | Units | Description | Reference |
|---|---|---|---|---|
| $\beta$ | 146.0 | 1 | Unit cost ratio, Eq. 3 | This study |
| $\Gamma^*_{25,p_0}$ | 4.332 | Pa | Photorespiratory compensation point, SC | Bernacchi et al. (2001) |
| $K_{c25}$ | 39.97 | Pa | MM coef. for $CO_2$, SC | Bernacchi et al. (2001) |
| $K_{o25}$ | 27480 | Pa | MM coef. for $O_2$, SC | Bernacchi et al. (2001) |
| $\Delta H_{\Gamma*}$ | 37830 | J mol$^{-1}$ | Activation energy for $\Gamma^*$ | Bernacchi et al. (2001) |
| $\Delta H_{Kc}$ | 79430 | J mol$^{-1}$ | Activation energy for $K_c$ | Bernacchi et al. (2001) |
| $\Delta H_{Ko}$ | 36380 | J mol$^{-1}$ | Activation energy for $K_o$ | Bernacchi et al. (2001) |
| $H_V$ | 71513 | J mol$^{-1}$ | Activation energy for $V_{\text{cmax}}$ | Kattge and Knorr (2007) |
| $H_d$ | 200000 | J mol$^{-1}$ | Deactivation energy for $V_{\text{cmax}}$ | Kattge and Knorr (2007) |
| $p_0$ | 101325 | Pa | Standard atmosphere | – |
| $g$ | 9.80665 | m s$^{-2}$ | Gravitation constant | – |
| $L$ | 0.0065 | K m$^{-2}$ | Adiabatic lapse rate | – |
| $R$ | 8.3145 | J mol$^{-1}$ K$^{-1}$ | Universal gas constant | – |
| $M_a$ | 28.963 | g mol$^{-1}$ | Molecular mass of dry air | – |
| $M_C$ | 12.0107 | g mol$^{-1}$ | Molecular mass of carbon | – |
| $a_S$ | 668.39 | J mol$^{-1}$ K$^{-1}$ | Intercept for entropy term in Eq. C6 | Kattge and Knorr (2007) |
| $b_S$ | 1.07 | J mol$^{-1}$ K$^{-2}$ | Slope for entropy term in Eq. C6 | Kattge and Knorr (2007) |

**Table A7.** Fixed parameters. 'SC' stands for 'at standard conditions' (25 °C, 101325 Pa). 'MM coef.' refers to 'Michaelis Menten coefficient'.





**Table A8.** Variables returned by the function `rpmodel()`. Variable names correspond to the named elements of the list returned by the `rpmodel()` function call. Symbols correspond to their use in this paper.

| Variable name | Symbol | Description | Units | Reference |
|---|---|---|---|---|
| ca | $c_a$ | Ambient $CO_2$ partial pressure | Pa | Sect. 2.1 |
| gammastar | $\Gamma^*$ | Photorespiratory compensation point | Pa | Sect. B1 |
| kmm | $K$ | Michaelis-Menten coefficient for photosynthesis | Pa | Sect. B3 |
| ns_star | $\eta^*$ | Change in the viscosity of water, relative to its value at 25 °C | unitless | Huber et al. (2009) |
| chi | $\chi$ | Ratio of leaf internal-to-ambient $CO_2$ | unitless | Sect. 2.1 |
| ci | $c_i$ | Leaf internal $CO_2$ partial pressure | Pa | Eq. E8 |
| lue | LUE | Light use efficiency | g C mol$^{-1}$ | Eq. 19 |
| mj | $m$ | $CO_2$ limitation factor for light-limited assimilation | unitless | Eq. 11 |
| mc | $m_C$ | $CO_2$ limitation factor for Rubisco-limited assimilation | unitless | Eq. 7 |
| gpp | GPP | Gross primary production | g C m$^{-2}$ d$^{-1}$ | Eqs. 2 and 19 |
| iwue | iWUE | Intrinsic water use efficiency | Pa | Eq. C2 |
| gs | $g_s$ | Stomatal conductance | mol C m$^{-2}$ d$^{-1}$ Pa$^{-1}$ | Sect. C1 |
| vcmax | $V_{\mathrm{cmax}}$ | Maximum rate of carboxylation | mol C m$^{-2}$ d$^{-1}$ | Eq. C4 |
| vcmax25 | $V_{\mathrm{cmax25}}$ | Maximum rate of carboxylation, normalised to 25 °C | mol C m$^{-2}$ d$^{-1}$ | Eq. C5 |
| rd | $R_d$ | Dark respiration | mol C m$^{-2}$ d$^{-1}$ | Eq. C11 |



*Author contributions.* B.D.S. designed the study, wrote the model code, conducted the analysis, and wrote the paper. H. W. developed the model and wrote the an initial version of the model description. N.G.S. developed the model and implemented model code. S.P.H. contributed to designing the study and writing the manuscript. T.K. contributed to study design, model implementation and manuscript writing. D.S. implemented the water holding capacity model. T.D. wrote an initial version of the model code and model documentation. I.C.P. developed the model and contributed to designing the study.

*Competing interests.* The authors have no competing interests.

*Acknowledgements.* B.D.S. was funded by ERC H2020-MSCA-IF-2015, grant number 701329. N.G.S. acknowledges support from Texas Tech University. T.F.K. acknowledges support from the Laboratory Directed Research and Development (LDRD) fund under the auspices of DOE, BER Office of Science at Lawrence Berkeley National Laboratory, and the NASA Terrestrial Ecology Program IDS Award NNH17AE86I. S.P.H. acknowledges support from the ERC-funded project GC 2.0 (Global Change 2.0: Unlocking the past for a clearer future, grant number 694481). I.C.P. acknowledges support from the ERC under the European Union's Horizon 2020 research and innovation programme (grant agreement no: 787203 REALM).



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
