# Peer review of "P-model v1.0: An optimality-based light use efficiency model for simulating ecosystem gross primary production"

_Geoscientific Model Development, 2019_

## Referee Comment (RC1) · Anonymous Referee #1 · 15 Sep 2019

I have a mixed feeling for this review. On the good side, this is incredibly well written. All figures and analysis are highly professional. On the other side, this manuscript degraded the elegance of optimality hypothesis.

I fully understand the original paper on optimality hypothesis (Han et al 2017) was not perfect. It had much room for improvements. But the way to improve in this manuscript is not attractive in my view with following reasons.

1) AET/PET was used for aridity index to consider drought effects. The authors used SPLASH model. If AET can be modeled so well, then GPP must be modeled well too as they are both tightly correlated via stomata conductance. Therefore, in my humble

opinion, bringing AET to consider drought effects in GPP estimates are logically odd. The key motivation of this study is to add soil stress function into P-model which leads better prediction of GPP, but that added soil moisture function appears decoupled from stomata conductance in the framework of optimality hypothesis. So in a physiological sense, it is not any more optimal model. Bringing stomata conductance from SPLASH would be one option although it is ugly... but the assumption of using AET/PET is that stomata conductance is correct.

2) There has been a series of papers that proposed global GPP maps with evaluations against fluxnet database. Many papers which were cited in this manuscript already evaluated model performance across scales from site level to the global land, daily to seasonal to annual scales. When I agreed to review this manuscript, I expected what would be global GPP, and how it varies in space and time from P-model. Site level evaluation for seasonal scale does not convince me about the overall performance of this model. In my past experience, I could match the modeled seasonal variations of GPP with fluxnet GPP extremely well; but in that case, global GPP values and interannual variation/trends were weird. I mean the authors should test the revised P model across different scales. Current evaluation is not enough.

3) The authors have incorporated an empirical soil moisture stress function to down-regulate LUEopt. I understand why the authors introduced soil moisture stress function after the 1st author's fantastic papers on drought and fLUE. However, I think the introduced soil module is too heavy given the elegance of optimality hypothesis. It is a typical soil bucket model which requires soil properties and rainfall. To scale up P-model globally, the key barrier will be this soil module- they are too uncertain and P-model will be coupled with a heavy hydrological model like SPLASH. We know microwave remote sensing based soil moisture only captures top soils.

4) The improved model still showed poor performance in capturing interannual variations of GPP. That's disappointed given the introduction of temperature and soil moisture terms.

5) Overall framework of revised P-model is almost identical to MODIS LUE model. MODIS GPP model downregulates LUEmax via temperature and VPD. Recent papers proposed a universal LUE max, or pixel based LUEopt that varies with time. That is the current status of MODIS GPP model. Then the revised P-model is almost following same direction; incorporating temperature and soil moisture to reduce LUEopt. Although the processes differ between two models in terms of f(temp) and f(water), overall philosophical framework appears very similar. That is the reason that I wrote "degradation of elegant P-model" in my general comments. If optimality hypothesis does not reflect temperature and water stress well, that indicates the optimality hypothesis is incorrect. Decoupling stomata conductance from added soil moisture function is a drawback in the framework of optimality. I would wish the authors incorporate temp/water effects into optimality theory in a more elegant way. The current way is too MODIS LUE style....

6) Current model evaluation is not enough. I strongly recommend testing the revised model at global scale across MODIS years. For example, Keenan et al (2017) showed recent increase of global GPP via P-model. Does the revised P-model still support this finding? Or does new modules of soil moisture and temperature reduce global GPP? I request this as P-model was already published so the authors may move many lines about original P model description to Appendix. The novelty of this model must be evaluation across diverse scales.

Only a few specific comments follow as the manusript is so well written. - L118: What was beta in Wang et al (2017a)? - L370: MODI -> MODIS

---

## Referee Comment (RC2) · Anonymous Referee #2 · 17 Sep 2019

Review for Stocker et al. GMD

Stocker et al present, calibrate, and evaluate a GPP model that is built off a previously published P-model. From the way the manuscript is structured it seems that the new additions to the P-model are the temperature dependence and the soil moisture stress (but this isn't clear from the abstract or the introduction). The authors should be commended for developing the model into an R package, using the FLUXNET2015 data in a way that recognizes the GPP is modeled product with considerable uncertainty (i.e., by analyzing multiple partitioning methods), and for providing throughout background for the modeling framework. Overall, the manuscript is well-written.

My recommendations for improvements are:

Be clearer about what is new to this version of the P-model. Based on the components of the model that are in the introduction vs. methods, I would assume that the temperature dependence and the soil moisture stress are the new components.

Since the modeling framework depends on the SPLASH model, more detail is needed about that model. How many parameters does it require and how where the SPLASH parameters determined? Any parameters that it requires should be added to Table A6.

It seems that the evaluation set included the calibration set plus additional sites. A more robust evaluation would use an independent set of sites for evaluation. I recommend presenting results for the set of independent sites to help understand how the model works out-of-sample.

The variable Iabs is not defined in the text (only in equations)

I recommend having the main text or the SI present the full model used predict GPP in its entirety. While I can piece it together across equations, a combined equation would help me understand the model in its complete form

The * in Table 1 is not defined (I think it is the footnote).

More detail is needed about the comparison to other GPP products described in the first paragraph of the discussion. Were they using the same evaluation dataset? Where their evaluations on out-of-sample sites or the same sites used in calibration? Also, it seems the P-model is only marginally better than the VPM and slightly worse that the annual MODIS GPP. Please provide more information and context for interpreting these comparisons.
* * *

---

## Author Comment (AC1) · 2 Dec 2019

**Response to Anonymous Referee #1**

We sincerely thank the reviewer for their effort and the very useful comments. We have revised our manuscript *P-model v1.0: An optimality-based light use efficiency model for simulating ecosystem gross primary production* and have addressed all points raised by the reviewer.

Below, we provide a point-by-point response to all the comments. Text by the reviewer is in blue and indented. Our response is in black. *New text is green, italic*. *Existing (unchanged) manuscript text is black italic*.

> I have a mixed feeling[s] for this review. On the good side, this is incredibly well written. All figures and analysis are highly professional. On the other side, this manuscript degraded the elegance of optimality hypothesis. I fully understand the original paper on optimality hypothesis (Han et al 2017) was not perfect. It had much room for improvements. But the way to improve in this manuscript is not attractive in my view with following reasons.

We thank the reviewer for the very positive assessment of the quality of text and figures. We also thank the reviewer for raising the point regarding the scope of the present manuscript and its relation to previous publications based on the same theoretical approach for modelling photosynthesis and its acclimation to the environment. We would like to highlight that the scope of the present manuscript is not to extend the theoretical approach described by (Prentice et al., 2014) and (Wang et al., 2017). We modified text in the introduction to clarify the purpose of the present manuscript:

*The purpose of this paper is (i) to provide a full documentation of the model implementation and reference for open-source software (rpmodelR package, https://stineb.github.io/rpmodel/); (ii) to provide an evaluation of model-predicted LUE and GPP against GPP derived from eddy covariance flux measurements (FLUXNET 2015 Tier 1 dataset); (iii) to apply this model for global-scale simulations and compare spatial patterns and global totals of simulated GPP with other estimates with global coverage; and (iv) to introduce a robust and pragmatic solution to resolving model bias under dry and cold conditions. With (iv) we do not aim at extending the theoretical basis for the P-model (Prentice et al., 2014; Wang et al., 2017), but to include environmental controls in the LUE model that serve to make the model applicable as a remote sensing data-driven GPP model for a wide range of conditions and vegetation types.*

More detailed responses, related to related points raised by reviewer #1, are given below.

> 1) AET/PET was used for aridity index to consider drought effects. The authors used SPLASH model. If AET can be modeled so well, then GPP must be modeled well too as they are both tightly correlated via stomatal conductance. Therefore, in my humble opinion, bringing AET to consider drought effects in GPP estimates are logically odd. The key motivation of this study is to add soil stress function into P-model which leads [to a] better prediction of GPP, but that added soil moisture function appears decoupled from stomata conductance in the framework of optimality hypothesis. So in a physiological sense, it is not any more optimal model. Bringing stomatal conductance from SPLASH would be one option although it is ugly... but the assumption of using AET/PET is that stomatal conductance is correct.

We think there are two points here. First, as a clarification, AET is not used directly to scale GPP in the model. Errors in simulated AET are thus not linearly translated into errors in simulated GPP. Instead, the annual mean fraction of daily AET/PET, simulated by the SPLASH model, is used to scale the *sensitivity* light use efficiency (LUE) to low soil moisture (Eq. 22). Furthermore, AET simulated by the SPLASH model is based on the Priestly-Taylor equation (Priestley and Taylor, 1972) for estimating potential evapotranspiration (PET), and thus assumes that PET is controlled by net radiation only,

independent of the vapour pressure deficit and stomatal conductance. In other words, this assumes a fully "decoupled" boundary layer (Jarvis, 1986). Therefore, "bringing stomatal conductance from SPLASH" is not actually possible.

The second point made here addresses the implementation of the soil moisture effects and its relation to the basic theory embodied by the P-model. With "basic theory" we mean principle of balancing the unit costs of transpiration and carboxylation following (Prentice et al., 2014) and the balancing of gains and costs associated with the maximum rate of electron transport, $J_{max}$ following (Wang et al., 2017). This theory is implemented by the equations described in Section 2 in our manuscript. The reviewer makes the point that the soil moisture stress function is not linked to the stomatal conductance predicted by the basic theory of the P-model. Point 5) (see below) is related to this point.

We fully agree with the reviewer as we explicitly state on l. 437:

*"[] the use of an empirical function is not consistent with the optimality approach that underlies the P-model."*

The potential for extending this theoretical framework and possible ways forward are discussed on subsequent lines (l. 522). We consider an inclusion of soil moisture effects into the theory underlying the P-model to be a very promising way forward and are actively working towards this goal. However, developing such an extension of the theoretical framework is a substantial piece of work as is demonstrated by the fact that similar efforts are currently being pursued by several research groups (Christoffersen et al., 2016; Mencuccini et al., 2019; Sperry et al., 2017; Wolf et al., 2016). In revising the manuscript text, we made sure to clarify that the empirical soil moisture stress function (Eqs. 21 and 22) is not to be understood as an extension of the optimality theory (Eqs. 3 and 14), but rather as a pragmatic solution to resolving known bias (see also our response above). E.g., on l. 69 we write:

*To resolve model biases under conditions of low soil moisture, (Stocker et al., 2019) further applied an empirical stress function to reduce LUE under dry soil conditions*

2) There has been a series of papers that proposed global GPP maps with evaluations against fluxnet database. Many papers which were cited in this manuscript already evaluated model performance across scales from site level to the global land, daily to seasonal to annual scales. When I agreed to review this manuscript, I expected what would be global GPP, and how it varies in space and time from P-model. Site level evaluation for seasonal scale does not convince me about the overall performance of this model. In my past experience, I could match the modeled seasonal variations of GPP with fluxnet GPP extremely well; but in that case, global GPP values and interannual variation/trends were weird. I mean the authors should test the revised P model across different scales. Current evaluation is not enough.

We now conducted global simulations with the P-model using the parameters from the site-level calibration (FULL setup) and we included a description of the respective model forcings, a presentation of results, and a comparison against other global GPP estimates and the spatial distribution of sun-induced fluorescence (SiF). We complemented text and figures in several instances. The most important changes/additions are:

Two new figures:

[revised manuscript text omitted]

We now realised that in the published *Discussions* manuscript, two short sections describing the results of the evaluations of 8-daily, interannual, and spatial variations were missing. Respective text was lost when formatting the document for *GMDD*. We duly apologize for this. In the revised manuscript, we added this text back to the manuscript. Figures and tables that show model performance for 8-daily, interannual, and spatial variations and that are referred to in added text had been included already in the published *Discussions* manuscript. Added text reads:

**4.1.1 8-day means**

*The P-model version ORG captures 69% of the variance in observed GPP with data aggregated to 8-day means (60'450 data points). Model performance both with respect to explained variance ($R^2$) and the RMSE is improved when additionally accounting for effects of temperature on the quantum yield efficiency (BRC, $R^2$ = 72%), and when additionally factoring in the empirical soil moisture stress function (FULL, $R^2$ = 75%, Fig. 2). The NULL model with temporally constant and spatially uniform LUE performed equally well as ORG, but is outperformed by model versions BRC and FULL. All performance statistics are given in Tables 3 and 4.*

*[...]*

**4.1.3 Spatial and annual variations**

*The $R^2$ for annual GPP simulated by the P-model setups ranges from 0.57 (ORG) to 0.70 (FULL). The NULL model achieves an $R^2$ of 0.58. Most of the explanatory power of the different models for predicting annual total GPP stems from their power in predicting between-site ("spatial") variations (Fig. 3, Tabs. 3 and 4). The $R^2$ for spatial variations ranges from 0.62 (ORG) to 0.70 (FULL), and 0.64 for the NULL model. In contrast, inter-annual variations (across years within a given site) are poorly simulated ($R^2$: 0.06-0.09 for P-model setups, and 0.04 for the NULL model). Interannual variations are generally much smaller than across site variations, which likely adds to the challenge of accurately capturing interannual variations. Interannual GPP variations are generally better captured at sites where the variability is high and in particular at dry sites.*

Taken together, we are providing a comprehensive evaluation, assessing 8-daily, annual, spatial, and seasonal variations; responses to droughts; and global GPP. In other words, we are not just evaluating seasonal variations as suggested by the reviewer. Temporal trends cannot currently be assessed using eddy-covariance data with confidence. This is mainly due to the (still) relatively short time series available from flux measurements (see below) and the relatively high interannual variability. Hence, we avoid comparing also temporal trends in global GPP.

[Figure]

**Fig. A** *Time series length in FLUXNET 2015 data for different sites in the Tier 1 set.*

3) The authors have incorporated an empirical soil moisture stress function to downregulate LUEopt. I understand why the authors introduced soil moisture stress function after the 1st author's fantastic papers on drought and fLUE. However, I think the introduced soil module is too heavy given the elegance of optimality hypothesis. It is a typical soil bucket model which requires soil properties and rainfall. To scale up P-model globally, the key barrier will be this soil module-they are too uncertain and P-model will be coupled with a heavy hydrological model like SPLASH. We know microwave remote sensing based soil moisture only captures top soils.

The soil water balance model implemented here is arguably the simplest form of such a model. It considers only one bucket (other models treat water movement across multiple soil layers which requires a computationally expensive numerical scheme), and simulates potential evapotranspiration following Priestly-Taylor (1972) (other models account for effects of boundary layer and surface resistance following the Penman-Monteith Equation). Any model that accounts for soil moisture effects must somehow treat soil moisture prognostically, unless it's prescribed from observations. We followed arguably the "lightest" possible prognostic soil moisture model, SPLASH. As the reviewer notes, observation-based soil moisture, e.g. based on microwave remote sensing (Al Bitar et al., 2016; Dorigo et al., 2017), are subject to the limitation that they only capture signals from the uppermost soil layers and may thus not be representative for plant water stress and of limited use for the present application.

4) The improved model still showed poor performance in capturing interannual variations of GPP. That's disappointed given the introduction of temperature and soil moisture terms.

We have added text in the discussion to put this result into context (l. 458).

*While seasonal and spatial variations in GPP are reliably simulated by the P-model, the model's performance in simulating interannual GPP variations is weaker. Similar results have been found from previous studies from both empirical (Richardson et al., 2007; Urbanski et al., 2007) and process model-based (Keenan et al., 2012) analyses. This is likely due to lagged effects of climate anomalies expressed through biotic responses (Biederman et al., 2016; Keenan et al., 2012; Richardson et al., 2007).*

5) Overall framework of revised P-model is almost identical to MODIS LUE model. MODIS GPP model downregulates LUEmax via temperature and VPD. Recent papers proposed a universal LUE max, or pixel based LUEopt that varies with time. That is the current status of MODIS GPP model. Then the revised P-model is almost following same direction; incorporating temperature and soil moisture to reduce LUEopt. Although the processes differ between two models in terms of f(temp) and f(water), overall philosophical framework appears very similar. That is the reason that I wrote "degradation of elegant P-model" in my general comments. If optimality hypothesis does not reflect temperature and water stress well, that indicates the optimality hypothesis is incorrect. Decoupling stomata conductance from added soil moisture function is a drawback in the framework of optimality. I would wish the authors incorporate temp/water effects into optimality theory in a more elegant way. The current way is too MODIS LUE style....

The model presented here differs from the MODIS GPP model (Running et al., 2004) in several fundamental ways. First, spatial variations in LUE are not prescribed, as is done by MODIS GPP based on biome-specific LUEopt values used in combination with a global biome classification, but are predicted based on the optimality principle (Eq. 3 in our manuscript) balancing costs associated with carbon gains and water losses. Second, the model presented here combines the enzyme kinetics for simulating photosynthesis described by the FvCB model with the Coordination Hypothesis to derive a

formulation for LUE that turns out to be linear with absorbed light. In other words, the LUE model concept emerges from the theory embodied in the model and is not pre-imposed.

As discussed above (see our response to point 1) the implementation of a simple water stress scalar was motivated by pragmatism. We modified text in the Discussion section to better elucidate mechanisms behind the "temperature stress" function applied here. Modified text reads:

*A reduction in the quantum yield efficiency arises from several mechanisms, including increased non-photochemical quenching, a reduction in chlorophyll and absorption by screening pigments (Adams et al., 2004; Ensminger et al., 2004; Huner et al., 1993; Oquist and Huner, 2003; Verhoeven, 2014). These adaptations serve to limit oxidative damage under high light and low temperature conditions, where an imbalance between electron supply and demand exists, arising from an imbalance between temperature-insensitive photochemical rates and temperature-sensitive biochemical rates. The reversion of these adaptations and resumption of the intrinsic quantum yield efficiency and photosynthesis requires sustained temperatures above a certain critical threshold (Rogers et al., 2017; Tanja et al., 2003) and exhibits a delay with respect to instantaneous air temperatures (Mäkelä et al., 2004; Pelkonen and Hari, 1980). Approaches accounting for a delayed resumption of photosynthesis after cold periods offer scope for further improvement of the P-model and may be included in global vegetation and Earth system models where this effect is currently not accounted for (Rogers et al., 2017; Tanja et al., 2003).*

We disagree that the apparent necessity to account for a water and temperature stress factor is indicative of the "optimality hypothesis [being] incorrect". Rather, it is *incomplete* (noting that every model is necessarily incomplete). We write on l. 521:

*The bias reduction associated with using an empirical soil moisture stress function hints at missing factors in the theoretical approach which rests on an assumed constancy of the unit costs of transpiration (a in Eq. 3).*

It might be a viable approach to include a cost associated with oxidative stress, governed by the imbalance between the photochemical and biochemical rates (Oquist and Huner, 2003). However, such an approach would have to account for the large diurnal variations in light availability. A reasonable alternative is to account for low-temperature stress by describing the phenomenon, as we do, rather than predicting its emergence based on photochemical and biochemical processes. We note that there are several other parameters and relationships in the photosynthesis model, including the light-response curve, that are empirical. The strength of our approach lies in our demonstration that the model can be effectively "scaled up" by taking account of acclimatory responses of model parameters.

As regards the soil moisture parameterization, this could (conceptually) be replaced by an extension of the optimality framework to include the additional costs of extracting water from dry soils. However, this will be a substantial project in itself, requiring (among other things) an optimality-based estimate of root-zone size. For the present, therefore, we find it useful to work with and present a hybrid model in which the response to low soil moisture is not optimality-based.

6) Current model evaluation is not enough. I strongly recommend testing the revised model at global scale across MODIS years. For example, Keenan et al (2017) showed recent increase of global GPP via P-model. Does the revised P-model still support this finding? Or does new modules of soil moisture and temperature reduce global GPP? I request this as P-model was already published so the authors may move many lines about original P model description to Appendix. The novelty of this model must be evaluation across diverse scales.

As discussed above in our response to the referee's point 2), we have added results and analyses of global P-model simulations. The version of the P-model used in Keenan et al. (2017) is different from the one applied here as described on lines 66-69. It did not include the $J_{max}$ limitation introduced by (Wang et al., 2017). We avoid presenting an evaluation of trends over the MODIS period (after February 2000) since we consider it too short for robust analyses. We have added text in the Discussion section to clarify these points.

*Due to the short period for which forcing data and outputs from comparable models are available, we did not analyse temporal trends in global GPP here. Analyses not shown here indicate that the introduction of the Jmax cost factor (not included, e.g., in (Keenan et al., 2016)) increases the sensitivity of modelled GPP to CO2. Further evaluation of model behaviour against data from CO2 manipulation experiments will be necessary before applying the model to simulate CO2-related trends.*

Only a few specific comments follow as the manuscript is so well written. - L118: What was beta in Wang et al (2017a)?

We added the values used by Wang et al. (2017) and in the present study:

*The unit cost ratio β has been estimated by Wang et al. (2017a) **to 240** based on global leaf δ13C data and a simplified version of the P-model (assumingΓ∗= 0 and neglecting the $J_{max}$ limitation). Here, we re-estimated β **to 146** based on the full version of the model using the same global leaf δ13C dataset.*

- L370: MODI -> MODIS

Done.

---

## Author Comment (AC2) · 2 Dec 2019

**Response to Anonymous Referee #2**

We sincerely thank the reviewer for her/his effort and the very useful comments. We have revised our manuscript *P-model v1.0: An optimality-based light use efficiency model for simulating ecosystem gross primary production* and have addressed all points raised by the reviewer.

Below, we provide a point-by-point response to all the comments. Text by the reviewer is in blue and indented. Our response is in black. *New text is green, italic*. *Existing (unchanged) manuscript text is black italic*.

> Stocker et al present, calibrate, and evaluate a GPP model that is built off a previously published P-model. From the way the manuscript is structured it seems that the new additions to the P-model are the temperature dependence and the soil moisture stress (but this isn't clear from the abstract or the introduction). The authors should be commended for developing the model into an R package, using the FLUXNET2015 data in a way that recognizes the GPP is modeled product with considerable uncertainty (i.e., by analyzing multiple partitioning methods), and for providing throughout background for the modeling framework. Overall, the manuscript is well-written.

We greatly appreciate this positive assessment and the recognition of the value of our open-access model implementation. In order to clarify the new additions and relation of the P-model to earlier publications early on, we added the following sentence to the abstract:

*[...] The model builds on the theory developed in Prentice et al. (2014) and Wang et al. (2017a) and is extended to include low temperature effects on the intrinsic quantum yield and an empirical soil moisture stress factor. [...]*

This is clarified further by the text in the introduction that we have added in response to referee 1:

*The purpose of this paper is [...] (iv) to introduce a robust and pragmatic solution to resolving model bias under dry and cold conditions.*

> My recommendations for improvements are:
>
> Be clearer about what is new to this version of the P-model. Based on the components of the model that are in the introduction vs. methods, I would assume that the temperature dependence and the soil moisture stress are the new components.

This is addressed by added text in the abstract and introduction as described above.

> Since the modeling framework depends on the SPLASH model, more detail is needed about that model. How many parameters does it require and how where the SPLASH parameters determined? Any parameters that it requires should be added to Table A6.

The SPLASH model was described in detail and all parameters defined in (Davis et al., 2017). The only difference to their model version is that we include a soil texture-dependent water holding capacity instead of a globally uniform value. This is now stated more explicitly. Modified text reads:

*Soil moisture (θ), AET, and PET are simulated using the SPLASH model (Davis et al., 2017), which treats soil water storage as a single bucket and calculates potential evapotranspiration based on Priestley and Taylor (1972).* **The only difference to the model version described by (Davis et al., 2017) is that** *we account here for a variable water holding capacity calculated based on soil texture and depth data from SoilGrids (Hengl et al., 2014). A detailed description of the applied empirical functions for calculating plant-available water holding capacity from texture data is given in Appendix D.*

Appendix D then provides a detailed description of how we derived water holding capacity values around the FLUXNET sites. We decided not to include further descriptions of the SPLASH model here in order to save space and to focus on what is implemented also in the *rpmodel* R package.

It seems that the evaluation set included the calibration set plus additional sites. A more robust evaluation would use an independent set of sites for evaluation. I recommend presenting results for the set of independent sites to help understand how the model works out-of-sample.

We thank the reviewer for this important point. Due to the small number of parameters that we calibrated simultaneously (1 for ORG and BRC, 3 for FULL setup) and due to the large amount of data from a wide variety of sites, the risk of over-fitting is small. This is confirmed by the additional calibration and evaluation we performed. Results show that the calibrated parameter values vary within around 1% across the entire set of out-of-bag calibrations. To clarify this point, we added content as follows.

We added text in the methods description in Section 3.3 Model calibration:

*To test the robustness of the calibration and evaluation metrics, we additionally performed out-of-sample calibrations for the FULL setup where the training set included data from all but one site. The test dataset, used to calculate R2 and RMSE, contained only data from that single left-out site.*

We added text and a new figure in a new results sub-section:

*4.1 Calibration results*

*The calibration of model parameters, done with data from all calibration sites simultaneously, yielded values that closely matched the means across parameter values derived from the out-of-sample calibrations (Fig. 2). This confirms the robustness of the calibration and a negligible degree of overfitting. Similarly for the evaluation metrics, the R2 and RMSE values reported from evaluations against data from all evaluation sites pooled yielded values that closely match the means across the out-of-sample evaluation metrics (each calculated with data from the single left-out site). This analysis also shows that the distribution of the evaluation metrics is skewed, with evaluations against a few sites indicating particularly relatively performance (R2 below 0.5 for ZM-Mon, AR-Vir, and FR-Pue), while the most frequent values indicate very good model performance (evaluations at 21 sites giving R2 values of above 0.8). Because the out-of-bag calibrations are computationally very demanding, we performed this analysis only for one setup (FULL) and below report evaluation metrics done with pooled data from all evaluation sites.*

[Figure]

**Figure 2**.*Out-of-sample calibration and evaluation results. (a-c) Distribution of parameter values from calibrations where data from one site was left out for each individual calibration. Parameter $\hat{a}_\theta$ and $\hat{b}_\theta$ are unitless. (d, e) Distribution of evaluation metrics calculated on data from the left-out site based on simulations with model parameters calibrated on all other sites' data. Solid red vertical lines represent the parameter values calibrated with data from all calibration pooled. These are the values reported in Tabs. 3 and 4. Dashed red lines represent the mean across values from out-of-bag calibrations and evaluations*

The variable $I_{abs}$ is not defined in the text (only in equations)

We added on line 118:

$I_{abs}$ is the amount of absorbed light and $\varphi_0$ is the intrinsic quantum yield efficiency.

I recommend having the main text or the SI present the full model used predict GPP in its entirety. While I can piece it together across equations, a combined equation would help me understand the model in its complete form

Unfortunately, to some degree, piecing it together is inevitable in view of the monstrous algebraic expression this would yield. To facilitate this point, though, we added a summary of the theory for GPP in Appendix F2:

*To sum up, the P-model calculates GPP as*

$$\text{GPP} = I_{\text{abs}} \, \varphi_0(T) \, \beta(\theta) \, m' \, M_C \, ,$$

*where*

$$m' = m \sqrt{1 - \left(\frac{c^*}{m}\right)^{2/3}} \, .$$

*and*

$$m = \frac{c_a - \Gamma^*}{c_a + 2\Gamma^* + 3\Gamma^* \sqrt{\frac{1.6\eta^* D}{\beta \, (K + \Gamma^*)}}} \, .$$

*$I_{abs}$ is the absorbed light (taken as fAPAR x PPFD, mol m-2), $\varphi_0$ is the temperature-dependent intrinsic quantum yield, $\beta(\theta)$ is the soil moisture stress factor, and $M_C$ is the molar mass of carbon (g mol-1).*

The * in Table 1 is not defined (I think it is the footnote).

We added the asterisk in the footnote.

More detail is needed about the comparison to other GPP products described in the first paragraph of the discussion. Were they using the same evaluation dataset? Where their evaluations on out-of-sample sites or the same sites used in calibration? Also, it seems the P-model is only marginally better than the VPM and slightly worse that the annual MODIS GPP. Please provide more information and context for interpreting these comparisons.

We write on line 457:

*Unfortunately, we cannot present a direct comparison between these models, based on data from identical dates and sites. A targeted model intercomparison may address this.*

In other words, we cannot state why the P-model performs better or worse than other comparable models. (Stocker et al., 2019) found that all the remote sensing-based models they investigated (most of which are referred to here as well), exhibited a systematic bias under drought conditions. The evaluation we provide here (Fig. 6) indicates that the P-model in its FULL setup largely resolves this issue. We further show that this resolution also leads to better predictions across all other scales investigated here (spatial, annual, seasonal, daily anomalies, see Tabs. 3 and 4). Due to missing published information on the other models' performance across all scales, we cannot provide a comparison at this level of detail.

**References**

Davis, T. W., Prentice, I. C., Stocker, B. D., Thomas, R. T., Whitley, R. J., Wang, H., Evans, B. J., Gallego-Sala, A. V., Sykes, M. T. and Cramer, W.: Simple process-led algorithms for simulating habitats (SPLASH v.1.0): robust indices of radiation, evapotranspiration and plant-available moisture, Geoscientific Model Development, 10(2), 689–708, 2017.

Hengl, T., de Jesus, J. M., MacMillan, R. A., Batjes, N. H., Heuvelink, G. B. M., Ribeiro, E., Samuel-Rosa, A., Kempen, B., Leenaars, J. G. B., Walsh, M. G. and Gonzalez, M. R.: SoilGrids1km--global soil information based on automated mapping, PLoS One, 9(8), e105992, 2014.

Stocker, B. D., Zscheischler, J., Keenan, T. F., Colin Prentice, I., Seneviratne, S. I. and Peñuelas, J.: Drought impacts on terrestrial primary production underestimated by satellite monitoring, Nature Geoscience, 12(4), 264–270, doi:10.1038/s41561-019-0318-6, 2019.

---

## Author Response (AR2)

Zurich, 3.2.2020

Dear Dr. Kala ,

Many thanks for handling the second round of revisions of our paper (gmd-2019-200 P-model v1.0: *An optimality-based light use efficiency model for simulating ecosystem gross primary production*).

I revised the code availability statement, now explicitly stating the version numbers and providing update references to Zenodo repositories (note that the citations *'Stocker [year]'* in brackets refers to respective Zenodo entries with their DOI given in the bibliography). I hope it's clear now that the *rpmodel* R package does implement all equations described in the paper, but for running the simulations, we used a Fortran version that implements the very same equations and gives identical results (as we have tested).

I have uploaded an updated PDF of the manuscript with the revised code availability statement here on the GMD MS overview. The revised text reads:

*The P-model is implemented as an R package (rpmodel) and available through CRAN and Zenodo (Stocker, 2019a). Results shown here correspond to rpmodel version v1.0.4. A documentation of the R package is available under https://stineb.github. io/rpmodel/. Both site-scale and global simulations shown here are done with the Fortran implementation of the P-model within the SOFUN modelling framework (version v1.2.0), available on Zenodo (Stocker, 2019b). Site-scale forcing data ingest and filtering, model calibration, and evaluation was done using the R package rsofun (version v1.0.wrap_sofun), available on Zenodo (Stocker, 2020b). Scripts that implement the workflow (repository eval_pmodel version v2) are available on Zenodo (Stocker, 2020a). Model outputs are available on Zenodo (Stocker, 2019c).*

I have also corrected a wrong description that came to our attention in the meantime on l. 186, replacing

*"Here θ is the plant-available soil water, expressed as a fraction of field capacity"*

with

*"Here θ is the plant-available soil water, expressed as a fraction of available water holding capacity"*

With kind regards and many thanks for your work as editor.

Sincerely, in the name of all co-authors,

Beni Stocker